# Disengaging spinal afferent nerve communication with the brain in live mice

Melinda A. Kyloh[1,6], Timothy J. Hibberd [1,6], Joel Castro[2,3], Andrea M. Harrington[2,3], Lee Travis [1], Kelsi N. Dodds [1], Lukasz Wiklendt[4], Stuart M. Brierley [2,3,5], Vladimir P. Zagorodnyuk [1] & Nick J. Spencer [1✉]

Our understanding of how abdominal organs (like the gut) communicate with the brain, via sensory nerves, has been limited by a lack of techniques to selectively activate or inhibit populations of spinal primary afferent neurons within dorsal root ganglia (DRG), of live animals. We report a survival surgery technique in mice, where select DRG are surgically removed (unilaterally or bilaterally), without interfering with other sensory or motor nerves. Using this approach, pain responses evoked by rectal distension were abolished by bilateral lumbosacral L5-S1 DRG removal, but not thoracolumbar T13-L1 DRG removal. However, animals lacking T13-L1 or L5-S1 DRG both showed reduced pain sensitivity to distal colonic distension. Removal of DRG led to selective loss of peripheral CGRP-expressing spinal afferent axons innervating visceral organs, arising from discrete spinal segments. This method thus allows spinal segment-specific determination of sensory pathway functions in conscious, free-to-move animals, without genetic modification.

[1] Visceral Neurophysiology Laboratory, College of Medicine and Public Health, Flinders Health and Medical Research Institute, Flinders University, Bedford Park, SA, Australia. [2] Visceral Pain Research Group, College of Medicine and Public Health, Flinders Health and Medical Research Institute, Flinders University, Bedford Park, SA, Australia. [3] Hopwood Centre for Neurobiology, Lifelong Health Theme, South Australian Health and Medical Research Institute, Adelaide, SA, Australia. [4] Discipline of Human Physiology, College of Medicine and Public Health, Flinders University, Bedford Park, SA, Australia. [5] School of Biomedicine, University of Adelaide, Adelaide, SA, Australia. [6] These authors contributed equally: Melinda A. Kyloh, Timothy J. Hibberd. ✉email: nicholas.spencer@flinders.edu.au

How abdominal organs, like the gastrointestinal tract, communicate via sensory nerves with the brain is of major interest to medical science. In recent years, considerable evidence has emerged to suggest that sensory nerve communication between the gut and brain plays a key role in health and disease[1,2]. A major limitation to our understanding of this field has been the lack of techniques to discriminate the functional roles of different types of sensory nerves that innervate the gut, and other abdominal organs. In vertebrates, it is well known that two distinct sensory nerve populations innervate abdominal organs, including the gut, which arise from either vagal[3,4], or spinal origin[5,6]. Compared with vagal afferents, our knowledge of the functional role of spinal afferent neurons in the body is particularly limited. This is because the dorsal root ganglia (DRG) that contain the nerve cell bodies of spinal afferents are spread across multiple different (thoracic, lumbar and sacral) spinal levels (Fig. 1). Furthermore, DRG are located deep under the vertebral foramina, making survival surgery intervention especially challenging. Dorsal root ganglia have been removed from mice with peripheral organs attached, but central pathways to the brain disconnected[7,8]. There has been a desperate need to determine the functional role

of specific populations of DRG in live animals, without imposing genetic modification.

All peripheral nerves that innervate abdominal organs, like the bladder, gut, or uterus, consist of mixed nerves, which are made up of both sensory (afferent) and motor (efferent) fibres that run in parallel, as they innervate their target organ[9,10]. Hence, the lack of techniques to selectively manipulate different populations of spinal afferents, without inadvertently interfering with vagal afferents, or any motor nerves, has hindered our understanding of how sensory nerves communicate between abdominal organs and the brain (like the gut–brain axis). In the past, nerve lesion studies have been applied to peripheral nerves close to the organ of interest. Unfortunately, these lesions inadvertently sever all afferent and efferent fibres, making interpretation of the physiological responses to the lesion unclear. Even more challenging has been how to discriminate functional roles of the different types of sensory neurons (vagal or spinal afferent)[6,11–13].

Here, we describe a survival surgery technique in mice, which allows us to ablate all sensory neurons in specific DRG of interest, either unilaterally or bilaterally, enabling long-term physiological or behavioural studies in conscious, free-to-move animals. The

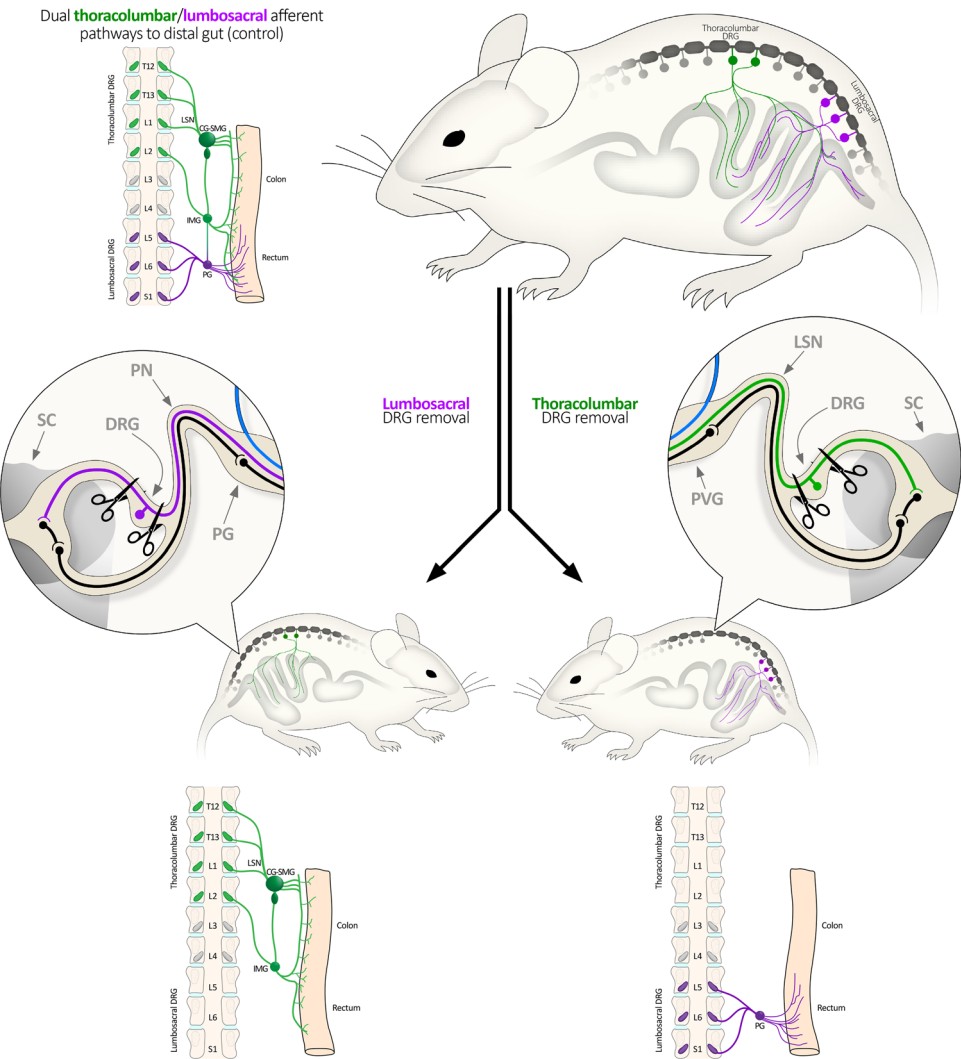

**Fig. 1 Schematic diagram showing types of DRG-lesioned mice created in this study.** Two major spinal afferent populations innervate the colon and rectum. They arise from the thoracolumbar DRG and the lumbosacral DRG. To study their functions, mice lacking either pathway were created by bilateral removal of the L5-S1 DRG (shown left), or the T13-L1 DRG (right). Unilateral removal of T13-L2 was used to study the uterine horns, allowing contralateral internal controls to be used. CG-SMG Celiac-superior mesenteric ganglia, IMG inferior mesenteric ganglion, LSN lumbar splanchnic nerve, PG pelvic ganglia, PN pelvic nerve, PVG prevertebral ganglion, DRG dorsal root ganglia, SC spinal cord.

approach described here reveals that by selective removal of one of the two distinct populations of DRG that innervate the colorectum[5,14–17], it is possible to ablate the pain signalling pathway and the visceromotor responses evoked by distension of specific regions of the colorectum. This approach will open the way for major investigations of either spinal afferents directly, or vagal afferent pathways (devoid of spinal afferents).

## Results

**Characteristics of DRG-lesioned mice**. We first investigated whether bilateral surgical removal of either lumbosacral (L5-S1; LS; Figs. 1–3), or thoracolumbar (T13-L1; TL; Figs. 1, 3 and 4) DRG would detectably affect the characteristics of mice related to food and water consumption, in vivo (Fig. 5a–l). Remarkably, neither LS nor TL-lesioned mice showed significant differences compared to sham control mice in daily measures of body-weight, food consumption, water consumption, dry faecal mass output, or the number of faecal pellets expelled (Data are summarised in Fig. 5a–l, and Supplementary Tables 1, 2). Likewise, the rate of ambulatory movement of mice at the same time points was not significantly different between either of the lesion and control groups (Fig. 5f, l; representative traces, Supplementary Fig. 1a, b).

**Visceral pain responses to colorectal distension following surgical removal of LS or TL DRG**. The colorectum features dual sensory innervation from distinct DRG neuron populations: those which arise primarily from the lumbosacral (L5-S1) DRG, and those from the thoracolumbar (T13-L1) DRG[5,14–18]. Using either LS or TL-lesioned mice, we tested whether these pathways contribute to pain signalling evoked by graded colorectal pressure distensions (20 to 80 mmHg) well into the noxious range. Distensions were applied to lesioned and sham control mice to evoke the visceromotor reflex (VMR)—a surrogate measure of evoked pain responses ($N = 14$, 8 and 7, sham controls, LS and TL-lesioned mice, respectively). The distension balloon (7 mm in length) was placed in two locations to evoke VMRs: the distal colon, ~16–23 mm rostral to the anus; and rectum, ~2.5–9.5 mm from the anus.

Distal colonic distensions evoked increasingly large bursts of firing activity in the abdominal muscle that graded with distension pressure in sham control animals (Fig. 5m–o). The absence of T13-L1 DRG in TL-lesioned mice attenuated the VMR, showing significantly decreased electromyography (EMG) firing compared to control mice from pressures of 40 mmHg onward (two-way, repeated measures ANOVA, Tukey post tests, see Fig. 5m–o). Notably, the compliance of the distal colon was no different across the three groups of mice, ($P = 0.074$, two-way ANOVA, main effect of surgery, $N = 14$, 7 and 8, sham controls, LS and TL-lesioned mice, respectively, Fig. 5p), suggesting the changes in VMR were due to the lack of afferents innervating the distal colon. Moreover, we found that LS lesioned mice also showed partial VMR reductions and were significantly reduced in response to 60 mmHg distensions, compared to control (two-way, repeated measures ANOVA, Tukey post tests, see Fig. 5m). Total VMR area under the curve (AUC; µV.s) of all distal colonic distensions were significantly reduced by the absence of T13-L1 ($P = 0.006$), but not L6-S1 DRG ($P = 0.080$) compared to sham control mice (one-way ANOVA Tukey post tests; see Fig. 5n). These results suggest both LS and TL DRG mediate pain signalling from the distal colon, with the TL DRG providing the most substantial contribution.

In sham control mice, graded distensions of the rectum similarly evoked VMRs ($N = 14$; Fig. 5q, r). In fact, VMRs in sham control mice did not differ between distal colonic and

rectal distension across the range of pressures tested ($P = 0.440$, $F (1,26) = 3.51$; two-way, repeated measures ANOVA, main effect of gut region, $N = 14$). The absence of L5-S1 DRG in LS lesioned mice abolished the VMR to rectal only distension across the complete range of pressure distensions (two-way, repeated measures ANOVA, Tukey post tests, see Fig. 5q–s). In contrast to distal colon distensions, the VMRs to rectal distension in TL-lesioned mice were not different to sham controls (see Fig. 5q, r) at any of the distension pressures tested (two-way, repeated measures ANOVA, Tukey post tests). Total VMR AUC of all rectal distensions were significantly reduced in LS lesioned mice compared to both sham controls ($P = 0.003$), and the TL-lesioned mice ($P = 0.001$). The TL-lesioned mice were not different to sham controls ($P = 0.598$; $t = 0.9761$, DF = 26; one-way, independent samples ANOVA Tukey post tests; see Fig. 5r). In both LS and TL-lesioned mice, the changes in VMR to distal colonic and rectal distension occurred despite normal VMRs to tail pinches, which were preserved in all groups of mice (Fig. 5o, s). Taken together, these results suggest pain signalling from the rectum cannot occur without sensory neurons in LS DRG, whilst both LS and TL DRG are important in the distal colonic region. Importantly, rectal compliance was also not significantly different between the three groups ($P = 0.165$, $F (1,23) = 1.95$; two-way, repeated measures ANOVA main effect of surgery, $N = 14$, 7 and 8, sham controls, LS and TL-lesioned mice, respectively, Fig. 5t). These results, together with those from distal colon above, indicate that the changes in the VMR to distension were caused by disruption of sensory signalling from DRG removal and not to any changes in viscoelastic properties of the gut wall.

To test the long-term stability of silenced pain signalling from DRG lesions, an additional cohort of LS lesioned animals and sham control animals were tested for VMRs to electrical stimulation 6 months after the lesion surgery ($N = 4$ animals per group). No detectable VMR was elicited in LS DRG-lesioned mice, following electrical stimulation of the rectum or bladder. These results showed statistical significance for bladder, but not rectum ($P = 0.029$ and 0.06 and, $t = 4.2$ and 3.6, DF = 6 and 5, bladder and rectum respectively, unpaired $t$ tests, adjusted for multiple comparisons; rectum VMR ($N = 4$ LS lesion and 3 sham); bladder VMR ($N = 4$ both groups); see Fig. 6). In the same mice, VMRs to tail, hindlimb and forelimb pinches were not significantly different from controls ($P = 0.973$, 0.627 and 0.973, $t = 0.2$, 1.2, 0.2, DF = 6, 6, 6, respectively unpaired $t$ tests, adjusted for multiple comparisons, $N = 4$).

**DRG lesion effects on cell activation marker expression in spinal cord**. To assess distension-evoked sensory signalling from the distal colon and rectum in more detail, the number of neurons activated within the spinal cord dorsal horn evoked by colorectal distension was compared between the experimental groups. Immunolabelling for neuronal activation marker pERK was quantified at T10–12, T13-L1 and L6-S1 spinal cord levels following noxious distension of the whole colorectum (~2.5–22.5 mm rostral to the anal opening) in sham control, LS- and TL-lesioned mice ($N = 5$ animals in each group). Consistent with VMR results, mice with removed LS DRG showed significantly reduced numbers of pERK+ neurons within the spinal dorsal horn compared to sham controls at the level of L6-S1, but not T10–12 or T13-L1 ($P < 0.0001$, $F (4,24) = 11$; two-way, repeated measures ANOVA interaction effect between type of surgery and vertebral level; see Fig. 7a–c). Conversely, mice with removed TL DRG showed significantly reduced numbers of pERK+ neurons across the dorsal horn at T13-L1, but not T10–12 or L6-S1 (see Fig. 7a–c). Thus, pain signalling from the

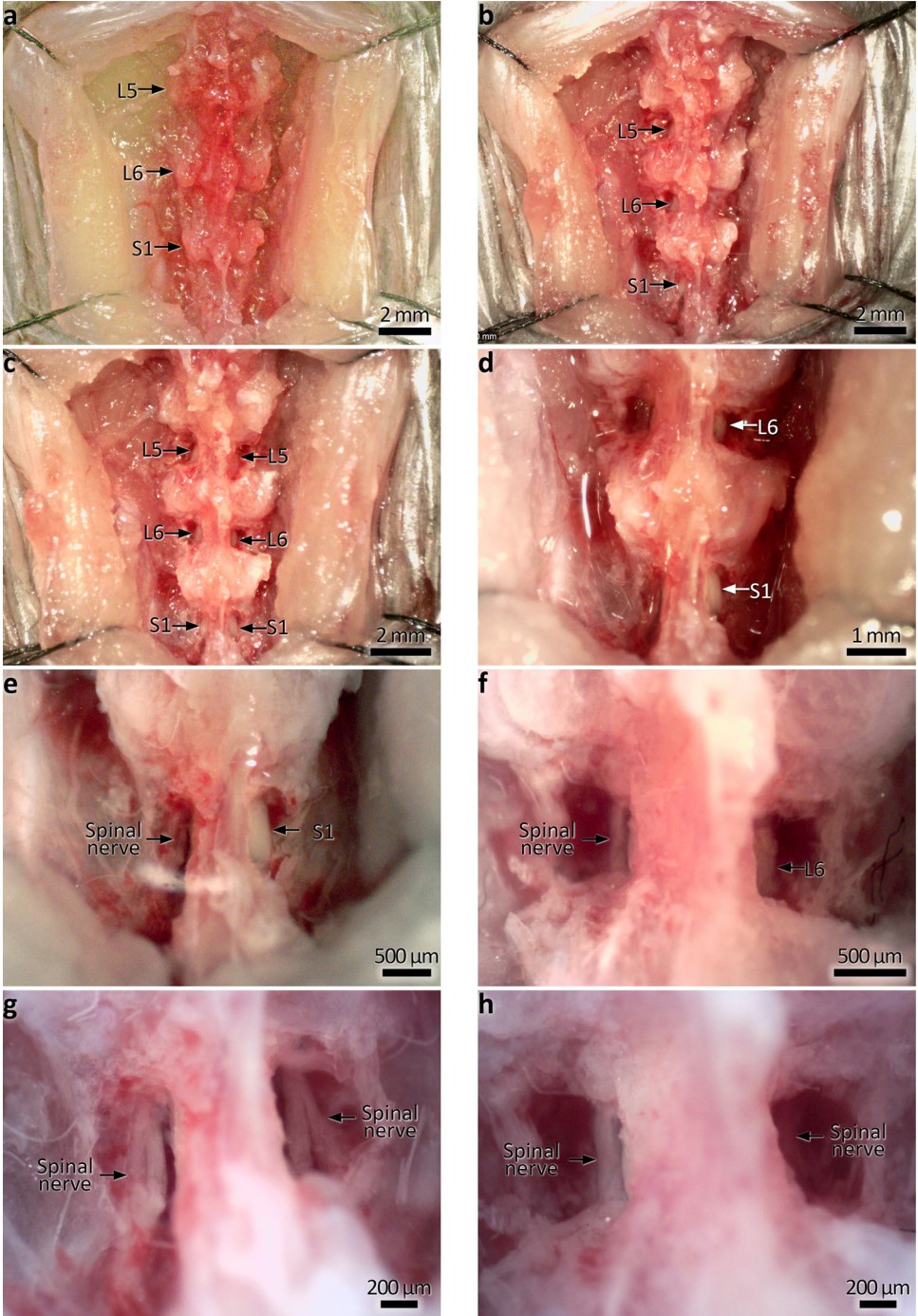

**Fig. 2 Stages of bilateral lumbosacral DRG removal. a** Midline section and retraction of the dorsal skin and musculature. DRG are not revealed, arrows indicate the vertebral levels; DRG lie at the junctions between vertebrae, approximately midway between the mammillary processes. **b** A single bank of DRG (L5-S1) are revealed by removing dorsal vertebral bone between the transverse processes. Arrows point to the exposed DRG. **c** Both banks of DRG exposed using the same method. Arrows point to the exposed DRG. **d** The DRG are carefully cut from the spinal nerves following exposure. This involves severing the dorsal root but not the ventral root (see Fig. 3a–d). In this higher magnification photo, L6 and S1 are removed on the left side; arrows indicate remaining intact DRG on the right side. The L6 DRG lies approximately parallel to the iliac crest of the hip bones. **e, f** Higher magnification photos of S1 and L6, respectively. Spinal nerves with DRG removed are indicated on the left side; intact DRG remain on the right side. **g, h** The same regions after bilateral DRG removal. An example of the section of the dorsal root prior to DRG removal ex vivo is shown in Fig. 3.

colorectum was selectively abolished in specific areas of the spinal cord by removal of selective sets of DRG. These results further support the idea that pain signalling from the colorectum is mediated by both LS and TL spinal sensory neurons. Analysis of distension-evoked pERK expression in major dorsal horn sub-regions are shown in Fig. 8a–f.

**Depletion of rectal spinal sensory axons following lumbosacral DRG lesion.** The neuropeptide calcitonin gene-related peptide (CGRP) occurs in the majority of spinal sensory axons in the gut wall[19,20]. To quantify the efficacy of DRG lesion, whole colons from LS lesioned and sham control mice were assessed for CGRP immunoreactivity 2 weeks after surgery, and in an additional

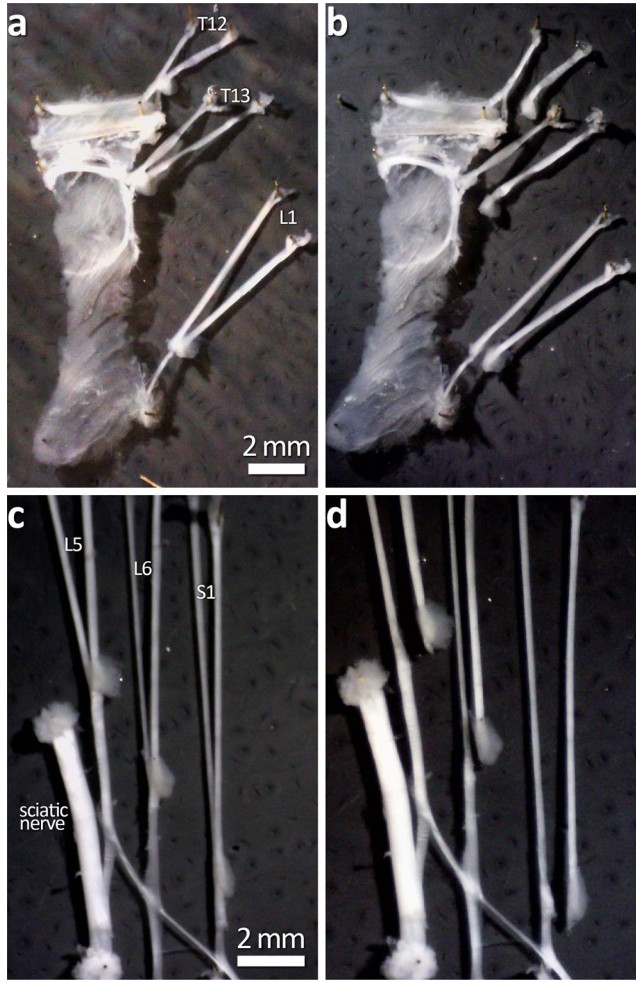

**Fig. 3 Ex vivo demonstration of dorsal root section prior to DRG removal. a, b** Dorsal root section of T12-L1. **c, d** Dorsal root section at L5-S1. This step is shown here in this way as the in vivo procedure gives little visual confirmation after the section of the root. Following this step, DRG are removed completely by cutting it away from the dorsal root, leaving the ventral root intact.

cohort 6 months after surgery. The rectal region from LS lesioned mice showed pronounced loss of CGRP immunoreactivity, compared to controls (see Fig. 9a, b and Supplementary Fig. 2a–h). At 2 weeks post surgery, the proportion of micrographs showing CGRP immunoreactivity was significantly reduced in rectum ($45 \pm 5$ vs. $31 \pm 5\%$ control vs. LS lesion, respectively, $P = 0.0008$, $t = 4.16$, DF = 22; Sidak post-test, two-way, repeated measures ANOVA, $N = 5$ animals each group), but not proximal colon ($44 \pm 3$ vs. $43 \pm 7\%$ control vs. LS lesion, respectively, $P = 0.0008$, $t = 4.16$, DF = 22; Sidak post-test, two-way, repeated measures ANOVA, N = 5 animals each group; Fig. 9b). This finding matches the data obtained in the pERK and VMR experiments directed to assess pain signalling and pain responses in vivo. The result was also similar at 6 months post surgery (Fig. 9b). These data indicate uncompensated loss of spinal afferent innervation of the distal colon and rectum occurred following LS lesion for up to 6 months post surgery. We compared CGRP immunoreactivity of the upper gut, where myenteric ganglionic density is much greater than the rectum and extrinsic vagal afferents still provide a prominent innervation. When we removed T10–T12 DRG from mice (which innervate the stomach), we were not able to detect a significant reduction in CGRP immunoreactivity ($N = 4$ animals in each group; Supplementary Fig. 3).

Consistent with the lack of change in daily pellet output (see Fig. 5d, e), LS lesioned colons showed normal neurogenic motor patterns, ex vivo (Fig. 10a–c). There was no significant effect on the frequency of the colonic motor complex (mean intervals: $225 \pm 74$ vs. $196 \pm 41$ s, control vs. LS lesion, respectively; $P = 0.522$, $t = 0.680$, DF = 6; independent samples 2 tailed $t$ test, $N = 4$ animals in each group; Fig. 10a).

**Depletion of uterine sensory axons following unilateral TL DRG removal.** Sensory innervation to the mouse uterus is predominantly supplied by spinal afferents arising from T13-L2 DRG, with a minority arising from L6-S1[21,22]. The lateralisation of this organ affords the opportunity for internal controls with unilateral DRG lesions. Thus, T13-L2 DRG unilateral lesions were performed before assessment of the ipsilateral and contralateral uterine horns for CGRP immunoreactivity. Examples of control (contralateral) and lesioned (ipsilateral) uterine CGRP immunoreactivity are shown in Fig. 9c. Quantified CGRP expression was significantly reduced across all analysed subregions of the uterine horn on the ipsilateral side compared to their contralateral controls in the same mice, with a more marked reduction observed in the oviduct end of the uterine horn (two-way, repeated measures ANOVA, $P = 0.002$, $F(1,4) = 52.13$; fixed effect of lesion side, $N = 5$ in each group; see Fig. 9d).

A separate series of experiments was performed to analyse CGRP immunoreactivity in a group of untreated control mice for comparison with data from lesioned animals ($N = 5$ animals each group). The independent control group showed no significant difference to either the ipsilateral or contralateral uterine horns in the lesioned animals, with values that were intermediate to the other groups. Statistically significant differences between the ipsilateral lesioned horn and its internal control, the contralateral uterine horn, remained at the oviduct end and mid region of the uterus, but not the cervical end (two-way, repeated measures ANOVA, $F(2,12) = 7.29$; fixed effect of uterine horn: ipsilateral lesion, contralateral lesion, and untreated control; Tukey post tests, $N = 5$ TL-lesioned animals and 5 non-surgical animals). This raises the possibility that TL DRG lesions may not only cause loss of CGRP-containing afferents in the target organ, but also induce compensatory changes in remaining spinal afferents that lead to an increase in CGRP immunoreactivity. These data are shown in Supplementary Fig. 4.

**Discussion**
We reveal a survival surgery technique which allows the functional role of spinal afferent neurons to be studied in conscious, free-to-move mice, without lesioning any other sensory or motor neurons. In an attempt to understand the physiological consequences of the loss of spinal afferents, previous studies have made peripheral nerve lesions within the abdominal cavity. An unfortunate consequence of peripheral nerve lesions is that they affect not only sensory nerves (spinal and vagal afferents), but also the extrinsic motor (efferent) nerves. Hence, lesions to peripheral nerves in vertebrates have led to vast uncertainty of the specific function of spinal afferent nerves. A major advantage of the technique presented here is that it is now possible to selectively ablate DRG of interest at specific spinal segments, either unilaterally or bilaterally, without incurring physiological deficits to other organs. This means that we can investigate the physiological function of specific organs of interest, following selective spinal afferent ablation from a particular division of the spinal cord. This has not been possible in the past and cannot be address using current chemogenetic techniques, which will affect all DRG spinal segments and requires genetically modified mice. The current method demonstrates that mice are viable and can be studied many

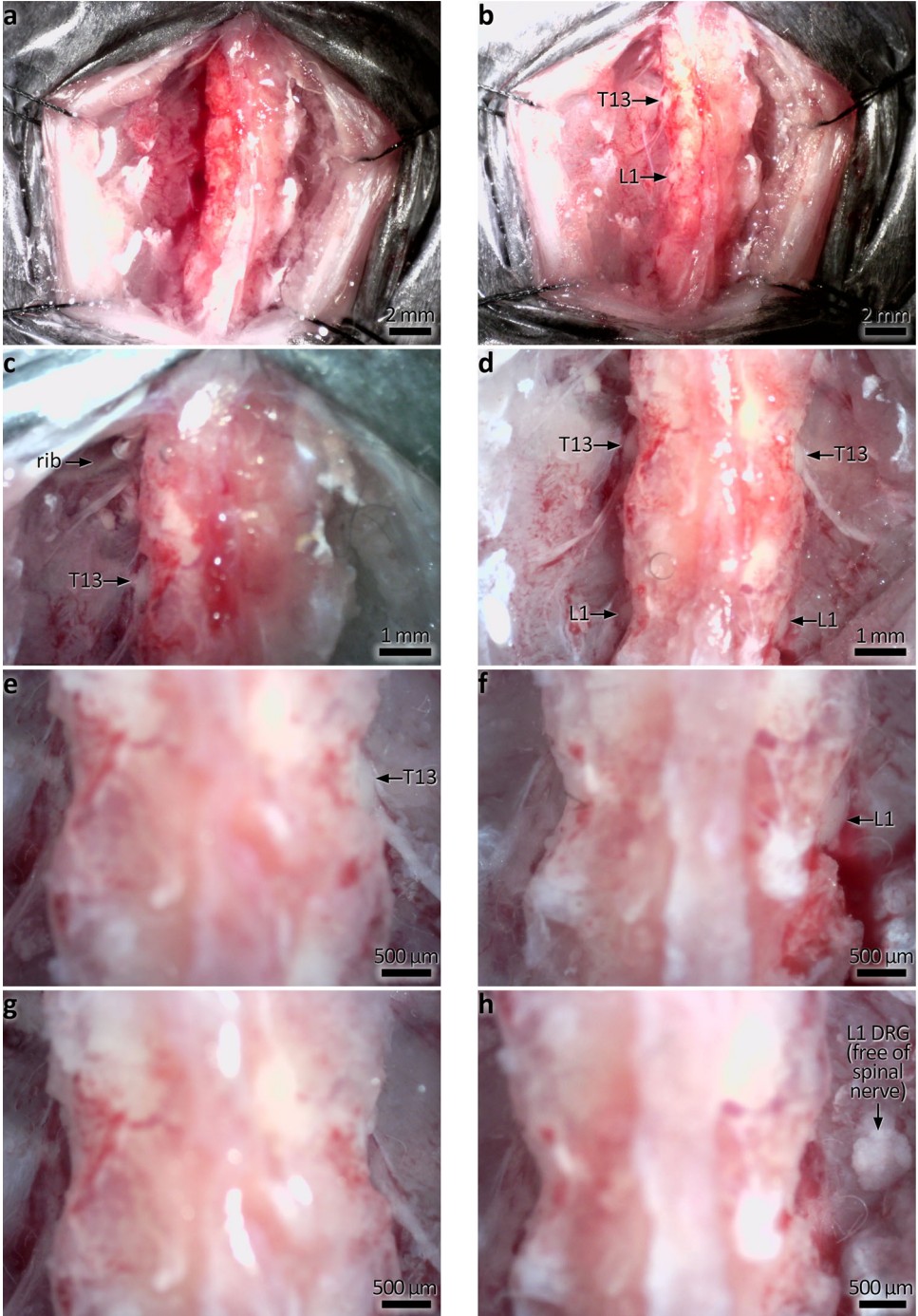

**Fig. 4 In vivo photographs showing DRG removal at T13-L1. a** The dorsal skin and musculature is cut either side of the spine. The most inferior ribs are used as a landmark; T13 DRG are located just inferior to the ribs. **b** Exposure of T13 and L1 DRG on the left side following partial removal of the vertebrae. Here the spinal nerves can be seen emerging from the DRG, which may be used as a guide before exposure. **c** Higher magnification, showing T13 DRG. The rib location is also indicated. **d** Bilateral exposure of T13 and L1 DRG. Prominent spinal nerves emerge from T13. **e**, **f** Unilateral removal of T13 and L1 DRG. **g**, **h** Bilateral removal of T13 and L1 DRG. Note the right L1 DRG shown floating free after removal (indicated by arrow).

months after DRG removal surgery, without regrowth of spinal afferents into the peripheral organs.

One of the great mysteries of visceral sensation in vertebrate animals, is why two different populations of DRG neurons innervate the same regions of visceral organs, like the gut, bladder or uterus (one in the thoracolumbar region and one in the lumbosacral region)[5,15–18,23–25]. The existence of these two discrete populations of DRG has led to an uncertainty as to which population mediates visceral pain signalling from specific abdominal organs. It was, therefore, of great interest to apply the technique described here to determine how pain responses elicited by colorectal distension would be affected by selective removal of each of the two distinct populations of DRG. We found the VMR evoked by noxious rectal distension was ablated by LS DRG removal, but not TL DRG removal. Moreover, mice with removed LS or TL DRG showed reduced pain responses to distension of the distal colon. Accordingly, a combined distension of the distal colon and rectum activated fewer neurons in the LS dorsal horn of the spinal

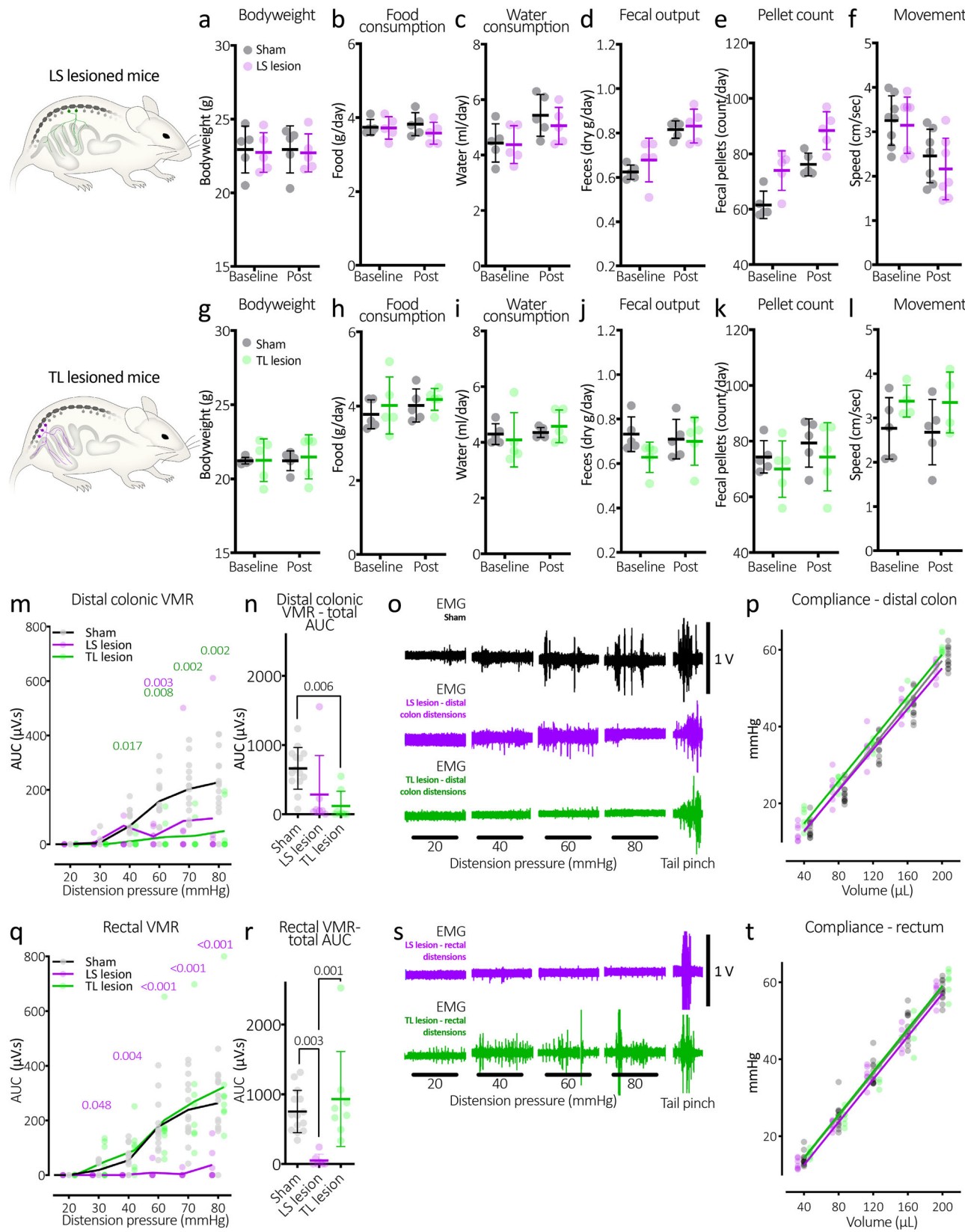

cord when LS DRG were removed. Notably, these very same animals have unchanged neuronal activation within the TL dorsal horn region of the spinal cord. Conversely, in TL DRG-lesioned animals the number of activated dorsal horn neurons was reduced specifically in the T13-L1 spinal levels and unchanged in the LS spinal regions. These results further confirmed the selectivity of the DRG lesioning technique, as the sensory signalling from non-lesioned DRG pathways remained intact. These findings are consistent with ex vivo afferent recording studies where pelvic (LS) but not splanchnic (TL) endings innervate the rectum, whereas both

**Fig. 5 Characteristics of LS and TL-lesioned mice and the abolition of the visceromotor reflex in LS lesioned mice. a–f** Bodyweight, food, water intake, faecal outputs and movement of LS lesioned mice. There were no significant changes in any parameter following the lesion surgery. **g–l** Characteristics of the TL-lesioned mice, also showing no significant changes following the lesion surgery. **m** Graph showing typical graded responses to distension of the distal colon in sham control mice (black line, grey markers). In contrast, TL-lesioned mice (green line and markers) had significantly reduced VMRs from 40 mmHg onward. *P* values from Tukey post-test comparisons with control values are shown in green above. LS lesioned mice also showed attenuated VMRs to distal colonic distension, showing a significant difference with control mice at 60 mmHg. **n** Total AUC of all distensions. TL-lesioned mice had significantly reduced AUC compared to control mice. *P* value refers to Tukey post-test. **o** Representative examples of VMRs to distal colon distensions in sham control, LS and TL-lesioned mice. Distension-evoked VMRs in control mice were similar in both rectal and distal colonic distensions. **p** The compliance of the distal colon was no different between the three groups of mice. **q** VMRs to rectal distension in LS lesion mice were largely abolished across the range of distension pressures. This suggests pain in the rectum is mediated by the lumbosacral sensory pathways. *P* values in purple text refer to Tukey post tests following two-way, repeated measures ANOVA: sham controls v LS lesion. **r** Total VMR AUC to rectal distensions was significantly lower in LS lesioned mice compared to both controls and TL-lesioned mice. *P* values in this figure refer to Tukey post tests. **s** Representative examples of VMRs to rectal distensions in LS and TL-lesioned mice. **t** The compliance of the rectum. This was no different between the three groups of mice. All error bars represent mean ± SD, individual markers represent individual animal averages.

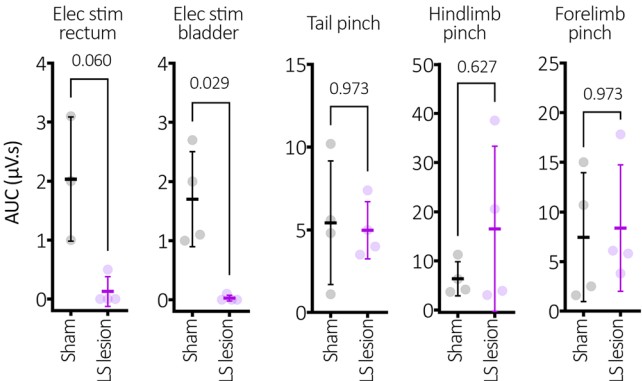

**Fig. 6 Long-term abolition of the visceromotor reflex.** Visceral insensitivity was maintained up to 6 months after surgery among LS lesioned mice, which showed a complete abolition of VMR to rectal and bladder electrical stimulation compared to sham mice. Tail and limb VMR responses showed no differences. *P* values refer to independent samples *t* tests (two-tailed). Statistically, there was no difference in VMR (*P* = 0.060) for rectal stimulation, while there was a significant (*P* = 0.029) for bladder stimulation. The lack of significance for rectal stimulation is likely attributable to the variability in sham EMG values, since VMRs were absent from rectum in DRG-removed mice. All error bars represent mean ± SD, individual markers represent individual animal averages.

splanchnic and pelvic afferents innervate the distal colon[26]. It is particularly noteworthy that 6 months after LS DRG removal, the VMR to electrical stimulation of the bladder or terminal colorectum was still ablated. This means that all afferents that contribute to the VMR, irrespective of mechanosensitivity, remain ablated months after surgical removal of this population of DRG. However, whilst VMR to electrical stimulation may be a more complete stimulus (since it will activate all afferents, i.e. mechanically sensitive and insensitive), this study did not test VMR to distension 6 months after DRG lesion, as was performed at 2 weeks after DRG removal. This could be considered a limitation of the present study. However, the important finding was that no VMR was activated 6 months after DRG removal, providing strong evidence that no functional afferents (that could underlie the VMR) were present. Likewise, the statistically significant ablation of the bladder VMR, but not rectal VMR at the 6-month time point is likely attributed to variability in EMG recordings from the sham mice, not DRG-lesioned mice (Fig. 6).

Our technique revealed that mice lacking multiple DRG live without overt loss of function for up to 6 months post-surgery (maximum duration tested). This means that the physiological

role of selective ablation of not only the DRG neurons, but the peripheral spinal afferent axons can be studied for long periods in conscious, free-to-move animals, without genetic modification. We also noted no changes in fecal pellet out, food intake, water intake, smooth muscle function and colonic motility after bilateral removal of LS or TL DRG. This technique can equally be applied in transgenic mice, so that chemogenetic or optogenetic tools can be combined with DRG removal.

Chemogenetic techniques have been used recently to distinguish between activation of spinal versus vagal afferents[13]. Whilst these approaches confer excellent genetic specificity for neuronal subtypes, a disadvantage in targeting spinal afferent neurons is its lack of anatomical specificity. Thus chemogenetic activation or inactivation cannot be confined to select vertebral levels. This can be overcome with our surgical technique, which may be applied unilaterally or bilaterally to ablate DRG at specific spinal segments, to ensure a select population of DRG are targeted. This is particularly useful for studying organs with bilateral symmetry, like the uterus, lungs, or kidney, where a single bank of DRG can be removed unilaterally, so that in the same animal one organ can be studied with spinal afferent innervation and the other without.

The surgical technique described here to expose DRG for their removal is the same technique that we use to inject DRG with neuronal tracers, for selective labelling of the axons and nerve endings of spinal afferents in visceral or somatic organs, down to single axon and nerve ending level[27,28]. Obviously, the difference between the two techniques is we do not remove the DRG, once we have injected DRGs with tracers. By injecting minute quantities of neuronal tracer into single DRG, it has been demonstrated we can readily identify the nerve endings that arise from a single DRG neuron[22,27,28]. This cannot be achieved using transgenic reporter or cre-induced expression of fluorescent reporters, which label large populations of axons whose spatial fields overlap extensively; see ref. 29.

We applied balloon distension to the terminal large intestine of conscious mice, at two different regions, one in the rectum, the other in the distal colon. Quantification of CGRP immunoreactivity showed a significant CGRP depletion in the rectal region, after LS DRG ablation. Different laboratories have used different terminology to describe the mouse terminal large intestine. We defined the rectal region of mouse colon as the region innervated predominantly by the rectal nerves (i.e. <16 mm of the anus). The distal colon was defined as the region (16–23 mm rostral to the anal sphincter). We discovered it was problematic trying to compare changes in extrinsic spinal afferent innervation along the length of large intestine, because even irrespective of any regional loss of spinal afferents, intrinsic ganglia also express CGRP[30,31] and the density of myenteric ganglia show a significant reduction along the length of large bowel[32]. In the rectal region, we were able to demonstrate a

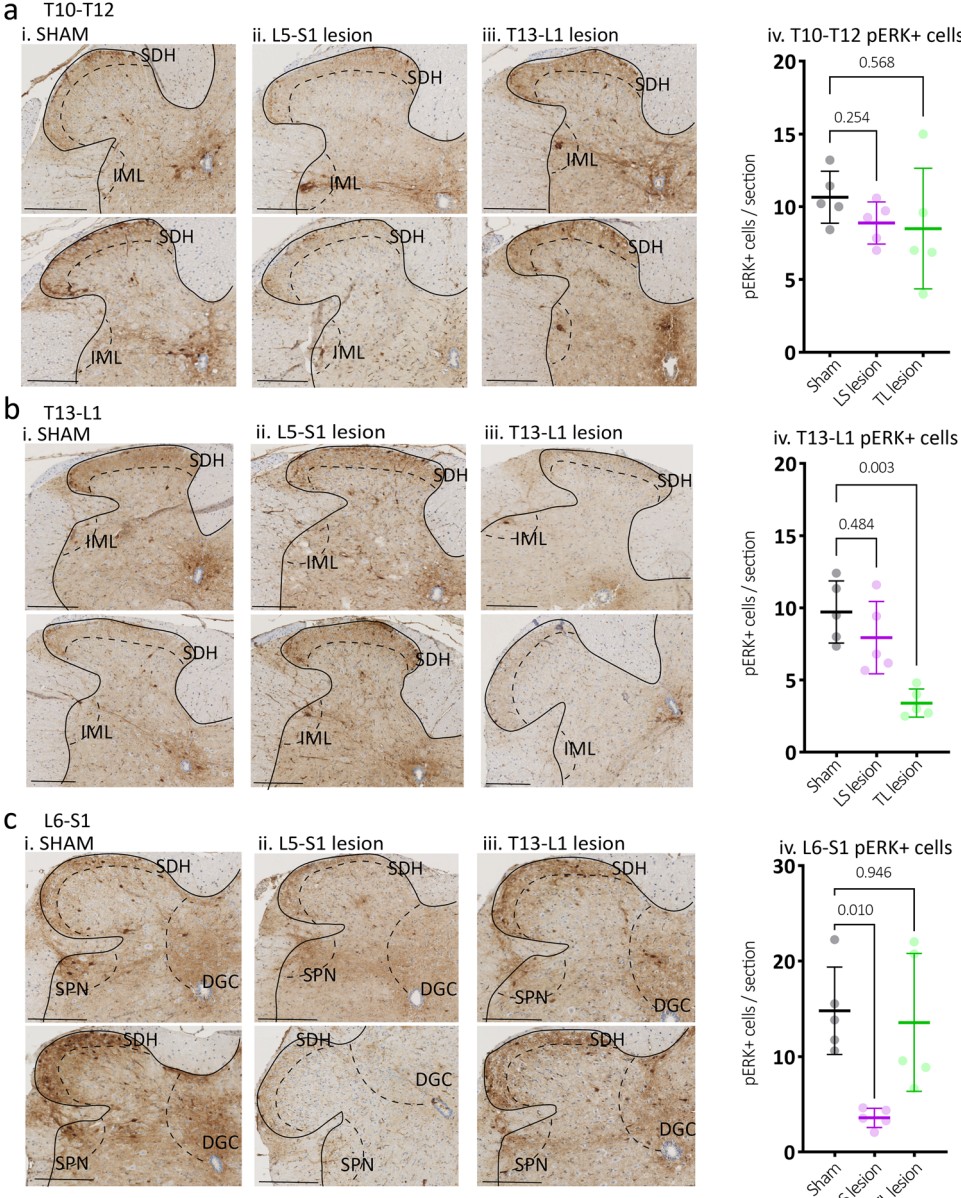

**Fig. 7 Expression of cell activation marker pERK in the dorsal horn of the spinal cord following noxious colorectal distension.** Representative images of pERK-immunoreactive (-IR) (dark brown) nerve cell bodies, evoked by noxious in vivo colorectal distension in cross-sections of (**a**) T10–T12, (**b**) T13-L1, and (**c**) L6-S1 spinal cord dorsal horn from (i) Sham, (ii) LS lesion and (iii) TL lesion mice. Calibration, 100 μm. In sham mice, pERK-IR neurons were observed in the superficial dorsal horn laminae I and II (SDH) at all spinal levels and below this in the in deep dorsal horn lamina (LIII-IV) in the L6-S1 spinal cord. pERK-IR neurons were present in the dorsal grey commissure (DGC), in the intermediolateral nuclei (IML) in thoracic T10–T12 sections and in the sacral parasympathetic nuclei (SPN) in sacral sections. iv The number of pERK-IR neurons evoked by in vivo colorectal distension was quantified within sections of (**a**) T10–T12, (**b**) T13-L1, (**c**) L6-S1 spinal cord dorsal horn from sham (grey markers), TL lesion (green markers) and LS lesion mice (magenta markers). The mean number of pERK-IR neurons/section in the T10–T12 dorsal horn did not differ between experimental groups ($N = 5$ per group). However, they were significantly reduced in the (**b**) T13-L1 dorsal horn of TL lesion mice, and in the (**c**) L6-S1 dorsal horn in LS lesioned mice relative to sham controls. P values refer to two-way, repeated measures ANOVA, Tukey post-hoc comparison tests. Individual data points represent the mean number of pERK-IR neurons per section per mouse. All error bars represent mean ± SD, individual markers represent individual animal averages.

significant reduction in CGRP immunoreactivity between sham and mice that underwent surgical ablation of L5-S1 DRG. This is because there are few myenteric ganglia in the rectum and this region lacks any vagal afferents (which also express CGRP). Indeed, in more proximal regions of gut, including the stomach, small bowel and upper part of colon, there is a prominent vagal afferent innervation and more intrinsic ganglia, which makes detection of changes in CGRP immunoreactivity (following DRG removal) unreliable (Supplementary Fig. 3).

In conclusion, application of the surgical technique outlined here will provide a major pathway for scientific enquiry, allowing discrimination of the functional role of spinal afferent neurons from vagal afferent neurons in the control of sensation from abdominal and somatic organs. It can also reveal the exact spinal pathways by which nociceptive information reaches the spinal cord and brain of live rodents. This approach not only allows selective investigation of spinal afferents from any possible role of vagal afferent signalling to the spinal cord and brain, but also

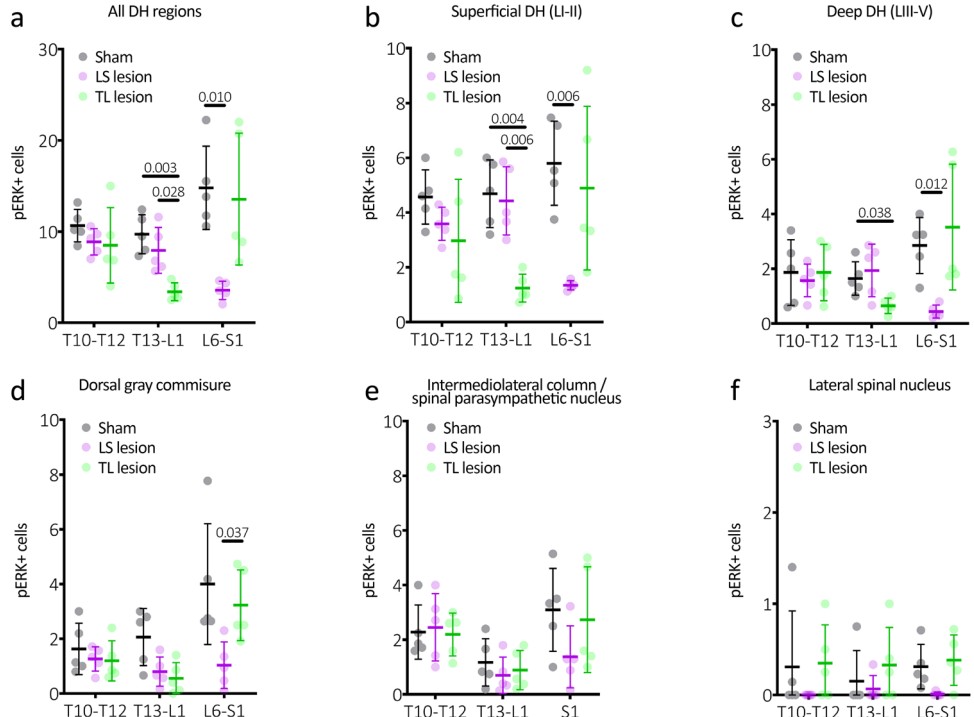

**Fig. 8 Subregion analysis of distension-evoked pERK expression in the spinal cord dorsal horn. a** Average counts from all dorsal horn regions. These data are identical to that shown in Fig. 5. **b** Counts from the superficial dorsal horn (LI–II), which typically showed the largest numbers of pERK-IR nerve cell bodies of all subregions analysed. Here it can be seen that the LS lesioned mice have significantly lower numbers of pERK-IR cells than the other groups at the level of the lesion. Interestingly, the TL-lesioned mice showed significantly fewer pERK-IR cells at the levels of T13-L1, compared to control. This suggests some involvement of the thoracolumbar pathways to distal colonic distension, despite lacking a detectable contribution to the VMR. **c** Counts from the deep dorsal horn (LIII-V). This shows a similar general pattern to that seen in the superficial layers, with lower cell counts. **d** Counts from the dorsal grey commissure, also showing a similar pattern to superficial and deep layers. Here, only counts at L6-S1 were significantly reduced in LS lesioned mice compared to TL-lesioned mice. **e** Counts from the intermediolateral column or spinal parasympathetic nucleus, showing no statistically significant differences between groups. **f** Counts from the lateral spinal nucleus were very low and showed no significant differences between groups. *P* values refer to two-way, repeated measures ANOVA, Tukey post-hoc comparison tests. Individual data points represent the mean number of pERK-IR neurons per section per mouse. All error bars represent mean ± SD, individual markers represent individual animal averages.

avoids any detrimental effects on motor nerve axons, which run alongside spinal and vagal afferents as they innervate their target peripheral organs.

## Methods

**Animals and surgery**. Procedures were approved by the Animal Welfare Committee of Flinders University (ethics approvals #861-13 and #933-16), and all protocols carried out in accordance with the National health and Medical Research Council (NHMRC) Australian code for the care and use of animal for scientific purposes (8th edition, 2013) and recommendations from the NHMRC Guidelines to promote the wellbeing of animals used for scientific purposes (2008). C57BL/6J mice of either sex (3–6 months of age) were anaesthetised using inhaled isoflurane; induced at 4% and then maintained at 1.5–2% in 1 L/min oxygen. Animals were positioned on a thermostat-controlled heat mat to maintain body temperature throughout the procedure (Adloheat, Pakenham, Vic, Australia). Before incision, animals were administered s.c. 0.05 mg/kg buprenorphine (Temvet). The dorsal surface was shaved and cleaned with 0.5% chlorhexidine and 70% alcohol swab (Briemar). An incision (~20 mm in length) was made along the dorsal midline and skeletal muscles retracted to expose the vertebral column. A partial laminotomy was performed to remove small fragments of vertebral bone overlying each dorsal root ganglion to expose the ganglion but not the dura or spinal cord (e.g. Figs. 2a–d, 4a–d). The dorsal nerve root was severed either side of the ganglion and the ganglion was removed entirely (see Figs. 2e–h, 3a–d and 4e–h). Images showing the progression of DRG removal are shown in Figs. 2 and 4. Following removal of DRG, the wound was irrigated with 0.5% Bupivicaine (Marcain, AstraZeneca) and muscle closed with individual 5.0 polyglycholic acid absorbable suture (Silverglide). Skin was closed with 6.0 Nylon non-absorbable suture (Silverglide) and the site cleaned with 0.5% chlorhexidine and 70% alcohol swab. Prior to withdrawal of anaesthesia, animals were administered a second s.c. dose of 0.05 mg/kg buprenorphine and s.c. antibiotics—100 mg/kg Ampicillin (Alphapharm) and 10 mg/kg Baytril (Bayer). Following withdrawal of anaesthesia, animals recovered on a heat mat with 1 L/min oxygen until fully mobile and then

returned to their home cage. Post operatively, animals received 0.1 mg/kg oral buprenorphine (Schering Plough) in Nutella (Ferrero) paste daily for 72 h.

For ex vivo and immunohistochemical studies, mice were euthanized using isoflurane inhalation overdose, followed by cervical dislocation.

**Sham procedure**. The identical procedure for sham mice was used as for DRG lesion mice, except that DRGs were not physically exposed in sham mice, nor lesions made to the dorsal nerve roots. An incision (~20 mm in length) was made along the dorsal midline and skeletal muscles retracted to expose the vertebral column. The identical anaesthesia protocol was used as for DRG-lesioned mice and animals were sutured using the same protocols.

**In vivo pain assessment: visceromotor response**. Abdominal EMG was used to measure the visceromotor response to colorectal distensions in conscious animals. At least 1 week after DRG lesion or sham surgery, mice were surgically implanted with EMG electrodes. Under inhaled isoflurane anaesthesia, the bare endings of two Teflon-coated stainless-steel wires (Advent Research Materials Ltd., Oxford, UK) were sutured into the right abdominal muscle. The wires were tunnelled subcutaneously to the base of the neck where they were exteriorised, coiled, and held in place at the base of the neck by sutures for future access. After surgery, mice received prophylactic antibiotic (Baytril; 5 mg/kg s.c.) and analgesic (buprenorphine; 0.04 mg/kg s.c.). Mice were housed individually and allowed to recover for at least 3 days before assessment of VMR. The day of VMR assessment corresponded to 12–15 days post DRG lesion or sham surgery. On the day of VMR assessment, mice were briefly anaesthetised using isoflurane. A lubricated balloon (0.7 cm length) was gently inserted into either the rectum (covering an area ~2.5–9.5 mm rostral to the anal opening) or the distal colon (covering an area of ~16–23 mm rostral to the anal opening) regions of the colon. The balloon catheter was secured to the base of the tail and connected to a barostat (Isobar 3, G&J Electronics, Willowdale, Ontario, Canada) for graded and pressure-controlled balloon distension. Mice were allowed to recover from anaesthesia in a restrainer, with dorsal access for 15 min prior to initiation of the distension sequence. Distensions were applied at 20–40–50–60–70–80 mmHg (20 s duration each) at 2 min intervals. Following the final distension, colonic compliance was assessed as

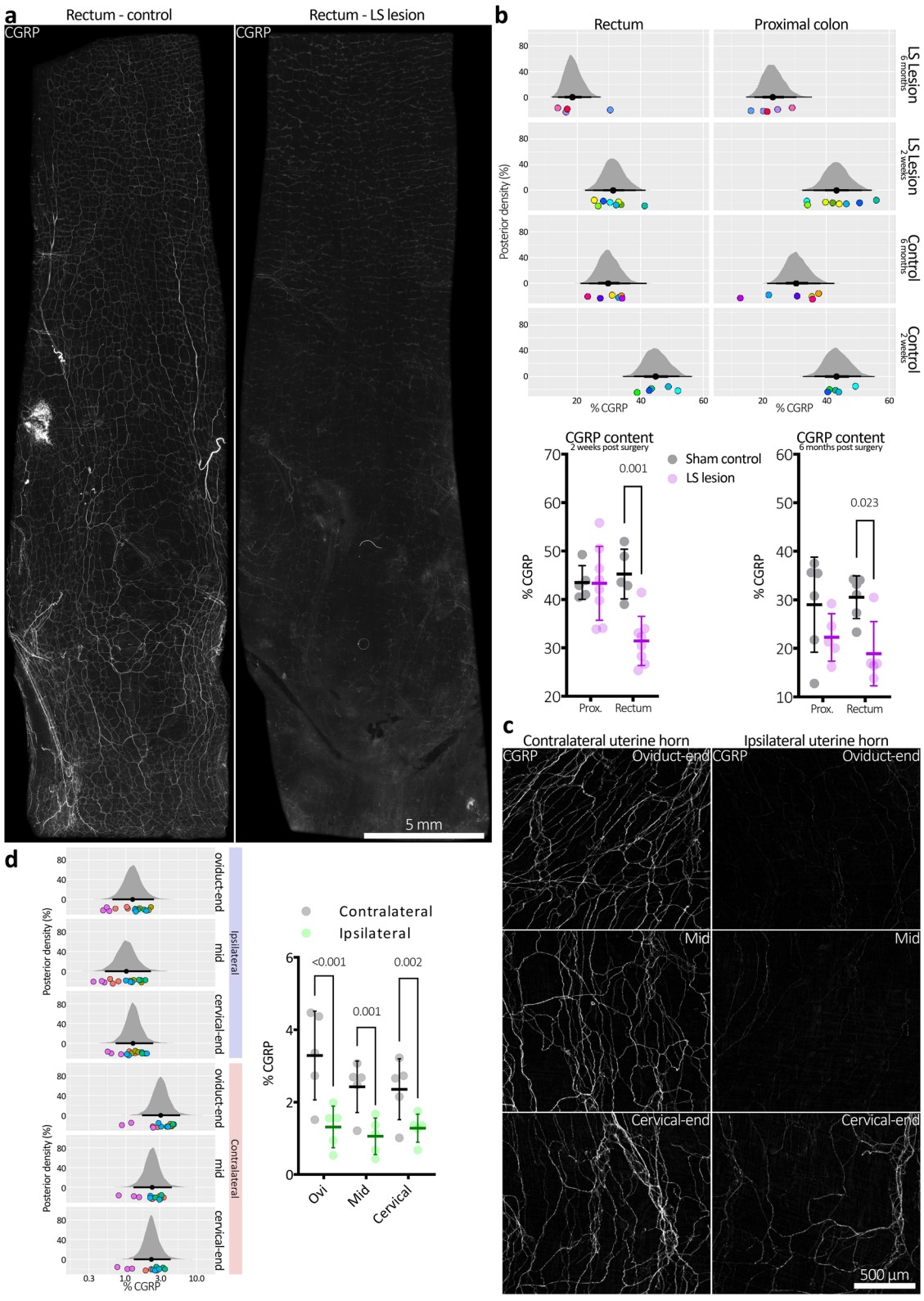

described below. The EMG electrodes were relayed to a data acquisition system and the signal was recorded (NL100AK headstage), amplified (NL104), filtered (NL 125/126, Neurolog, Digitimer Ltd., bandpass 50–5000 Hz), and digitised (CED 1401, Cambridge Electronic Design, Cambridge, UK) for off-line analysis using Spike2 software (Cambridge Electronic Design). The analogue EMG signal was rectified and integrated. To quantify the magnitude of the VMR at each distension pressure, the area under the curve (AUC) during the distension (20 s) was corrected for baseline activity (AUC pre-distension, 20 s). We also calculated the total AUC, which is the summation of data points across all distension pressures for each individual animal.

VMRs to electrical stimulation of rectum and bladder 6 months after surgery were performed by anaesthetising with pentobarbital sodium (200–300 µl of 6 mg/ml, ~40–60 mg/kg); depth of anaesthesia was assessed by lack of response to hindlimb or tail pinch. Electromyographic electrodes were placed into the left external oblique muscle and a reference electrode placed in the quadriceps muscle of the opposing leg. EMG recordings were acquired at 20 kHz and recorded on a PC running LabChart 7 Pro software and high pass filtered (100 Hz). To calculate VMR the analogue EMG signal was rectified and AUC (in µV.s) was calculated in LabChart 7 Pro (ADInstruments, Australia). A pair of

**Fig. 9 Depletion of spinal afferent neuropeptide CGRP, after DRG removal. a** CGRP immunofluorescence in distal colons from control (sham) mice and from mice lacking bilateral L5-S1 DRG, both 2 weeks after surgery. Intensely CGRP-immunofluorescent axons are prominent in mouse colorectum, characteristic of spinal afferent nerves. Less intense CGRP immunofluorescence occurs in a subclass of enteric neuron. Whilst intense CGRP labelling was abolished by the LS lesion, CGRP was still detectable in enteric neurons. This is consistent with selective ablation of spinal afferent neurons from the colon in LS lesioned animals. **b** Summary of CGRP immunohistochemical results in both short (2-week post surgery) and long term (6 months post) mice. Here, posterior densities (vertical axis) represent probability distributions of the proportion CGRP immunoreactivity. The black dot below the centre of a distribution represents the mean and the horizontal black lines either side of the dot, from thickest to thinnest, represent the 50, 95 and 99.5% confidence intervals. CGRP immunoreactivity is quantified as the proportion (%) positive pixels per immunofluorescence micrograph. The results show significant reductions in CGRP immunofluorescence in the distal, but not proximal colon in both types of lesioned mice compared to their controls. This suggests the ablation of spinal afferent neurons is stable over long periods of time in the rectum. The spatial pattern of CGRP depletion is consistent with the known distribution of LS afferent neurons that enter the colorectum via pelvic/rectal nerve pathways. Coloured dots below the posterior densities represent individual animal averages. Identically coloured dots in horizontally adjacent graphs represent values from the same animal. The same data are presented below the panel of posterior densities as standard graphs, showing mean ± SD and individual markers for each animal average. **c** Representative micrographs showing CGRP immunofluorescence in uterine horn following unilateral removal of T13-L2 DRG. **d** Quantitative summary of CGRP immunofluorescence in the uterine horns. Ipsilateral CGRP density was significantly decreased relative to the contralateral horn in all subregions sampled. Posterior densities have the same definitions as described in (b) and coloured dots represent replicate measures (identical colours represent values from the same animal). The same data are also presented adjacent the panel of posterior densities as standard graphs, showing mean ± SD and individual markers for each animal average.

stainless-steel stimulating electrodes insulated to within ~2 mm of the tip were inserted 4–6 mm into the colorectum of mice for rectal stimulations, and then placed on the bladder surface following partial midline laparotomy for bladder stimulation. Square single pulse electrical stimuli were generated using a Grass SD9 stimulator unit (60 V, 0.5 ms pulse width). Following experiments, mice were euthanised by an overdose of pentobarbital sodium, followed by cervical dislocation.

**Colonic compliance.** Immediately after conscious VMR assessment, colonic compliance was assessed by applying graded volumes (40–200 µL, 20 s duration each) to the balloon in the distal colon and rectum, while recording the corresponding colorectal pressure, as described above. Mice were euthanised by isoflurane inhalation overdose, followed by cervical dislocation after the measurement of colonic compliance.

**CGRP immunohistochemistry.** Full-length colon and whole uteri (female mice) were removed from euthanized mice that were ~2 weeks (colon or uterus) or 6 months post-surgery (colon only), cut along their respective mesenteric or mesometrial borders, and pinned epithelial (mucosal/endometrial) side uppermost as flat-sheet preparations in Sylgard-lined (Dow-Corning #3097358-1004; Midland, MI) Petri dishes containing phosphate-buffered saline (PBS; 0.1 M). Preparations were then fixed overnight in 4% paraformaldehyde (in PBS; pH 7.2). Mucosa or endometrium, respectively, were dissected free of the underlying muscular layers, cleared with dimethyl sulfoxide, and blocked for 1 h with 10% normal horse serum (NHS; Life Technologies Gibco #16050-122; Scoresby, Australia). Preparations were then incubated in primary antibody (rabbit anti-CGRP; 1:2000 dilution from neat antiserum in 10% NHS; Peninsula Laboratories International Inc. #T-4032; San Carlos, CA) for 2 days. Finally, tissues were incubated in secondary antibody (donkey anti-rabbit Cy3; 7.5 µg/ml; Jackson ImmunoResearch Laboratories Inc. #711-165-152; West Grove, PA) for 4 h, before mounting serosal or perimetrial side uppermost, respectively, on glass slides with 100% carbonate-buffered glycerol (pH 8.6). All antibodies and block solutions were diluted with PBS containing 0.1% sodium azide and PBS washes were performed between all antibody incubation steps.

Slides were viewed with an Olympus IX71 epifluorescence microscope (Shinjuku-ku, Tokyo, Japan) using appropriate laser wavelengths. Images were captured as TIFF files at ×4 magnification with a CoolSNAP™ camera (Roper Scientific, Tucson, AZ,USA) and AnalySIS Image 5.0 computer software (Olympus-SIS,Münster, Germany). Background autofluorescence in each image was normalised using the *Subtract Background* feature of Fiji Image J 1.52p software (www.fiji.sc; RRID: SCR_002285)[33] with the rolling ball radius set to 50.0 pixels. CGRP density (%), denoting the proportion of immunoreactive structures, was determined using code written in PyCharm Community Edition 2019.3.4 software (www.jetbrains.com/pycharm; RRID: SCR_018221). In brief, the code analyzed TIFF images in 8-bit grayscale and calculated the average overall pixel intensity of each image, where total black pixels = 0% (immunoreactivity absent) and total white pixels = 100% (immunoreactivity present). Average CGRP (pixel) intensity values were statistically analyzed with the brms library in R[34]. A Bayesian generalised linear model in the Beta family with a logit link function was used to compare CGRP expression intensities between rectal region of sham or DRG-transected animals, or uterine horns ipsilateral and contralateral to DRG transection. For analysis of CGRP, we defined the rectal region as the region predominantly innervated by the rectal nerves (within 16 mm of the anal sphincter). The brms R library[34] was used to

perform the multi-level model fit. The Beta location (mu) and scale (phi) parameters had equivalent formulations given by Eqs. 1–3.

$$\text{For rectum}: ''\text{surgery} * \text{period} * \text{tissue} + (1|p|\text{animal})'' \qquad (1)$$

Where surgery was either sham or lesion, period was either 6 months or 2 weeks, and tissue was either proximal colon or rectum.

$$\text{For uterine horn}: ''\text{group} * \text{location} + (\text{group} * \text{location}|p|\text{animal})'' \qquad (2)$$

Where group was ipsilateral or contralateral, and location was oviduct-end or mid or cervical-end.

$$\text{For stomach}: ''\text{surgery} * \text{region} + (\text{region}|p|\text{animal})'' \qquad (3)$$

Where surgery was either sham or lesion, and region was Les (lower oesophageal sphincter), Fundus, Corpus, Antrum, or Duodenum.

The "animal" component grouped measures from the same animal under a common random effect. The priors for the fixed-effect location and scale coefficients were set to standard normal, and the correlation matrix prior was set to LKJ(2) Priors for all other parameters were left as brms defaults. There were 8 chains of 1000 warmup iterations and 1000 post-warmup iterations, totalling 8000 useable posterior samples. Diagnostics revealed no divergences and Rhat was close to 1.

To control for oestrous-related changes in uterine volume, myometrial thickness was assessed in cryostat-sectioned tissues by manually placing a calibrated line over the appropriate uterine wall layers in Fiji Image J software and executing the *Measure* function. Average thicknesses were calculated from three measurements within a given tissue segment, which were then used to divide CGRP density values for each corresponding uterine region by a thickness ratio, defined as the average regional thickness divided by average of all thickness values.

**Food and water consumption, faecal output and ambulation.** Mice were individually housed for 24 h monitoring of food and water consumption and faecal output. Measurements were taken at a similar time daily, for 5 consecutive days in the week preceding DRG lesion or sham surgery for baseline measurements, and then again for 5 consecutive days starting on day 6 post surgery. Food and water were weighed, and bedding was collected and replaced daily. Faecal pellets were retrieved from bedding, counted, and then dehydrated for 24 h at 60 °C (BioChef Kalahari 10; Byron Bay, NSW, Australia) before weighting.

To quantify ambulation, mice were placed in a circular arena (20 cm diameter) containing regular bedding materials, and video recorded for a 15 min duration. Recordings were done on the day after daily monitoring was complete before and after surgery. Mice were discriminated and tracked in videos using Ethovision XT software (Noldus, Seattle, WA).

**Ex vivo colonic motility.** The full-length colon was removed and placed in a Petri dish filled with carbogen-gassed (95% $O_2$/5% $CO_2$) Krebs solution (~30–35 °C; in $10^{-3}$ M concentrations: NaCl 118; KCl 4.7, $NaH_2PO_4$ 1; $NaHCO_3$ 25; $MgCl_2$ 1.2; D-Glucose 11; $CaCl_2$ 2.5) and gently flushed of content before transferring to an organ bath for mechanical recordings. The organ bath (volume ~50 ml) was continuously superfused with Krebs solution at ~5 ml/min (36 °C). A stainless-steel tube (diameter 1 mm) placed through the lumen was fitted at each end into L-shaped barbed plastic connectors that were fixed to organ bath base. The oral and anal ends of colon were tied over the barbed connectors with fine suture thread. Four stainless-steel hooks (250 µm diameter) were threaded through the wall of the proximal, mid-proximal, mid-distal and

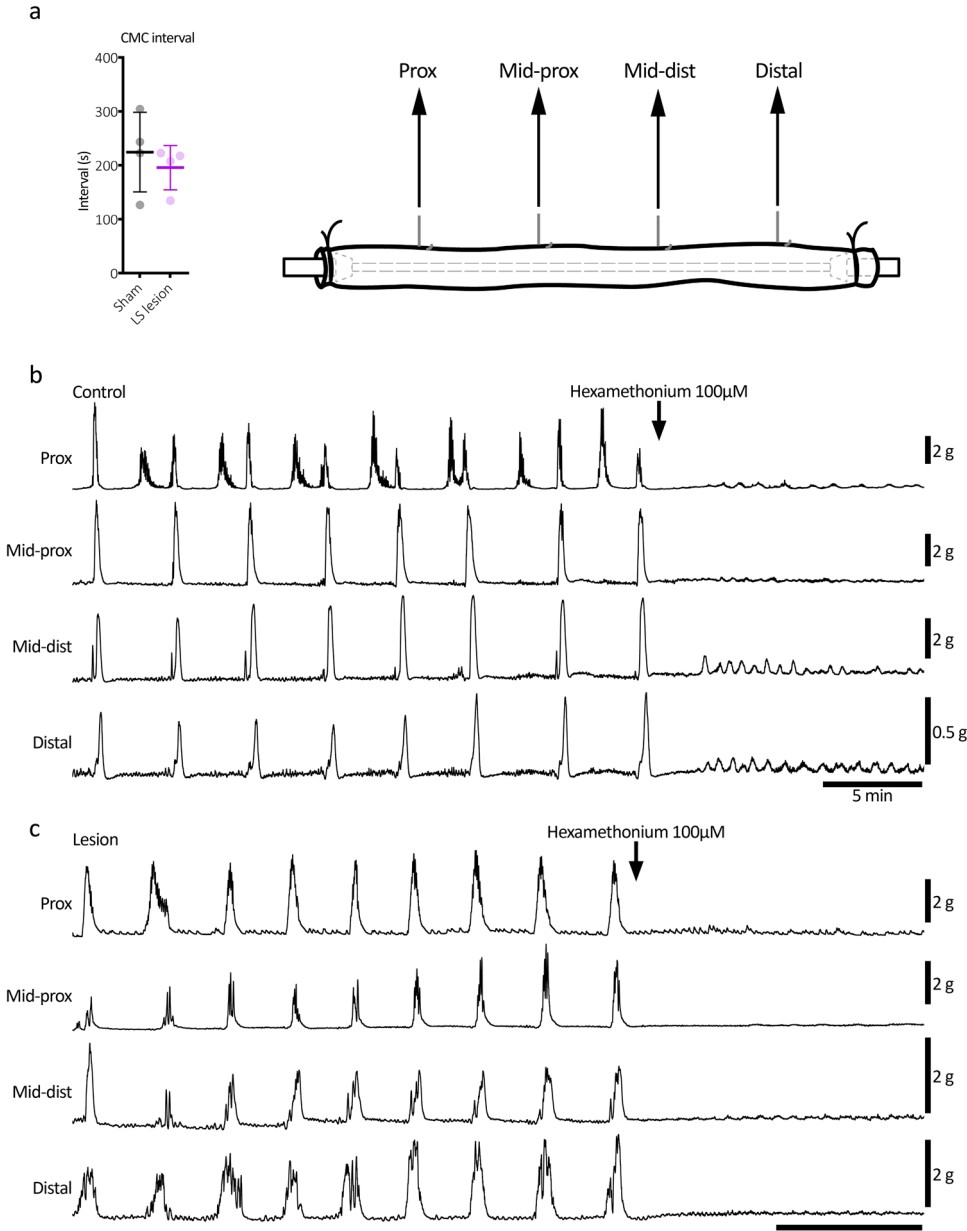

**Fig. 10 Ex vivo colonic motility after lumbosacral DRG lesion. a** The removal of L5-S1 DRG had no significant effect on the frequency of the colonic motor complex ex vivo (mean intervals: 225 ± 74 vs. 196 ± 41 s, control vs. LS lesion, respectively; $P = 0.522$, $t = 0.680$, DF = 6; independent samples 2 tailed $t$ test, $N = 4$ in each group). Error bars represent mean ± SD, individual markers represent individual animal averages. **b**, **c** Representative mechanical recordings from control and LS lesioned mice, respectively. Hexamethonium promptly abolished high amplitude contractions in both examples, confirming their dependence on nicotinic transmission within the enteric nervous system.

distal colon (Fig. 10). A fine suture thread connected each hook to an isometric force transducer (Grass FT-03C; Grass, Quincy, MA, USA), which was set to a basal tension of 0.5 g to record ongoing mechanical activity for 60 min at 1 kHz sampling rate (PowerLab 16/35, LabChart 8, ADInstruments, NSW, Australia).

**Identification of spinal cord dorsal horn neuron activation evoked by in vivo colorectal distension**. 10–14 days following DRG lesion or sham surgery, mice underwent in vivo acute noxious colorectal distension with a 2 cm latex balloon followed by immediate transcardial perfuse fixation. Regions of the thoracolumbar (T10-L1) and lumbosacral (L5-S2) spinal cord were then removed and underwent processing for immunolabelling for the neuronal activation marker phosphorylated MAP kinase ERK 1/2 (pERK). Mice were briefly anesthetised, during which a 2 cm long balloon catheter was inserted into the colorectum and secured to the tail. Mice were removed from isoflurane induction chamber and as they regained consciousness the balloon catheter was distended to 80 mmHg (5 × 10 s distension with 5 s deflation) via a syringe attached to a sphygmomanometer pressure gauge. After

the final distension, mice were given an overdose of pentobarbitone i.p. (Lethabarb, Virbac, Australia) and within 5 min underwent transcardial perfuse fixation with warmed 0.1 M phosphate buffer followed by ice-cold 4% paraformaldehyde in 0.1 M phosphate buffer.

After complete perfusion, the spinal cord between vertebrae T8–T10 (inclusive thoracolumbar spinal cord levels T10–T12), T11–T13 (inclusive thoracolumbar spinal cord levels T13-L1) and vertebra L1-L5 (inclusive lumbosacral spinal cord levels L5-S1) was removed and post-fixed in 4% paraformaldehyde in 0.1 M phosphate buffer at 4 °C for 18 to 20 h. Spinal cords were transferred to 30% sucrose in 0.1 M phosphate buffer then into 50% OCT/30% sucrose in 0.1 M phosphate buffer prior to freezing in 100% OCT using liquid nitrogen cooled isopentane. Frozen spinal cord samples were cryosectioned on a cryostat (Leica CM 1950) with 10 μm thick cross-sections of T10–T12, T13-L1 and L6-S1 spinal cord placed onto slides (InstrumeC Uberfrost Printer Slides) and stored at −200 °C prior to processing for pERK immunolabelling. Spinal cord levels were confirmed using the Allen Spinal Cord Atlas available from https://mousespinal.brain-map.org.

**pERK immunohistochemistry**. Phosphorylated MAP kinase ERK 1/2 (pERK) immunolabelling was performed on sections from the different experimental groups using a DAKO Omnis auto-stainer. The primary antibody (pERK 1/2, 1:800, MAB4370, Cell Signalling Technology, Genesearch, Qld) diluted in antibody Diluent (S0809, Agilent DAKO, Santa Clara, CA) was detected with 3,3′-Diaminobenzidine (DAB)/horseradish peroxidase (HRP) secondary antibody staining. Non-specific binding of secondary antibodies was blocked with Serum-Free Protein Block (X0909, Agilent DAKO). Tissue sections were pre-incubated with primary antisera for 1 h, washed and incubated in Envision FLEX Peroxidase-blocking Reagent (GV823, Agilent DAKO), followed by Envision FLEX HRP Polymer (GV823, Agilent DAKO) for HRP binding. Sections were then washed in wash buffer (GC807, DAKO Omnis, Agilent) before a 10 min incubation in EnVision FLEX Substrate Working Solution (DAB).

All slides were imaged using a NanoZoomer slide scanner (Hamamatsu, Japan) with a ×40 objective. At the time of scanning, the images produced were assigned a random number that de-identifies their experimental group. The scanned images were then opened and viewed using free NDPview2 software (https://www.hamamatsu.com/jp/en/product/type/U12388-01/index.html). The images were not manipulated in any way.

**Neuronal counts and data analysis**. The number of pERK-positive neurons was counted in one side of the dorsal horn in randomly selected sections of T10–T12, T13-L1 and L6-S1 spinal cord from scanned images opened in digital pathology viewing software QuPath 0.1.2. The mean number of pERK-immunoreactive dorsal horn neurons/section was obtained from 5 to 10 sections/mouse. The mean number ± SD of pERK immunoreactivity/section in the entire dorsal horn (inclusive of laminae I–V, dorsal grey commissure (DGC), intermediolateral nuclei (IML) in thoracic spinal sections, and the sacral parasympathetic nuclei (SPN) in the sacral sections), in the superficial dorsal horn (inclusive of laminae I–II) and the DGC was compared between experimental groups.

**Statistics and reproducibility**. Statistical testing was performed on animal averages. Replicate values represent measurements arising from independent animals. Statistical analyses were performed by parametric one-way or two-way, repeated measures ANOVA followed by appropriate post-hoc comparison tests, and by student's $t$ test for independent samples using the Holm-Šídák method to adjust $P$ values for multiple comparisons where appropriate (Prism 9 analysis software). Statistical differences were considered significant if $P < 0.05$. Results are expressed as mean ± standard deviation. Upper case $N$ always indicates the number of animals used in a set of experiments.

**Reporting summary**. Further information on research design is available in the Nature Research Reporting Summary linked to this article.

## Data availability

The datasets generated during and/or analysed during the current study are available from the corresponding author on reasonable request. Numerical source data supporting all figures and tables in this paper are available for immediate download from FigShare: https://doi.org/10.6084/m9.figshare.20444682.v1.

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

## Acknowledgements

Work was supported by a National Health and Medical Research Council of Australia (NHMRC) Project Grant (APP1156427 to N.J.S, S.M.B, V.Z, A.H), NHMRC Ideas grant (APP1184546 to V.Z.), NHMRC Investigator Grant (APP2008727 to S.M.B.) and an Australian Research Council (ARC) Discovery Projects (DP180101395 to A.M.H and S.M.B.) and DP220100070 to NJS. Figure 1 incorporates DRG and laboratory mouse diagrams that were adapted from[35] and Gwilz (2013)[36], respectively.

## Author contributions

M.A.K., L.T., T.J.H., J.C., A.M.H., N.J.S. and K.N.D. performed experiments. T.J.H., J.C., A.M.H., N.J.S., L.W. and K.N.D. drafted the paper. S.M.B., V.P.Z., J.C. and N.J.S. contributed to experimental design. All authors contributed to data analysis/interpretation and edited the paper.

## Competing interests

The authors declare no competing interests.
