## [Peer Review File · Communications Biology]

Reviewers' comments:

Reviewer #1 (Remarks to the Author):

The authors use a novel surgical technique to remove dorsal root ganglia from different levels of the mouse spinal cord, thus ablating specific subsets of spinal sensory neurons that (e.g., lumbosacral (LS) or thoracolumbar (TL)) innervate various organs. This approach led to the major conclusion that LS sensory neurons mediate pain from the rectum, whereas TL sensory neurons mediate pain from the distal colon. Additional immunohistochemistry experiments were used to support these findings, but were inconsistent/incomplete compared to their visceromotor response experiments (see comments below). Importantly, the surgical approach did not alter feeding behavior, weight, fecal output, or locomotion. The authors also state that sensory neurons remain depleted after 6 months with lasting reductions in visceromotor responses in colorectum and bladder (although the p value for colorectum VMR is not significant, $p=0.06$). Several comments and suggestions to increase the scientific impact and significance of the paper are provided below:

- In the introduction the authors suggest that there is a lack of techniques to manipulate spinal sensory neurons versus motor neurons, however this has been done in ex vivo preparations. While theirs is the first in vivo approach, the previously published ex vivo study may be worth noting (Smith-Edwards and Najjar et al., 2019. Gastroenterology PMID: 31075226).
- Another paper by the same group (Meerschaert et al., 2020. J Neurons PMID: 32817244) used retrograde tracing and single cell transcriptomics to look at differences in innervation of the colon and bladder as well as molecular expression patterns in LS, TL, and vagal sensory neurons. The authors should discuss how their findings compare and contrast to this previously published study.
- The authors need to include a description of the sham surgery in the methods section to increase the reproducibility of the work.
- In Figure 4, panel M: the magenta line for the VMR in LS-lesioned mice in response to distal colon distension does not appear to represent the average of the data points at pressures 70 and 80 mmHg; the visible magenta dots are all below the line, so it is unclear what data points are causing the average to be higher? Is it possible that there were some very high values that are out of the range of the y axis?
- Regarding Figure 5: For consistency and more convincing results, the authors should have used the same balloon (the 7mm one) and distended the rectum and distal colon to compare pERK in various spinal regions with and without DRG lesioning. These experiments would have provided additional and more significant data on whether 'labeled line' pathways continue in the spinal cord or could provide evidence regarding the extent of spinal integration from different DRG neurons and receptive fields.
- Also related to consistency and more convincing results, the authors should perform sham, LS, and TL lesion surgeries when quantifying all of their endpoints, especially considering their interesting findings in Figure 4. For example, only LS lesion was done to quantify the loss of CGRP endings in the colon (Figure 6) as well as in the 6-month VMR experiments (Supplemental Figure 3).
- Further, it is unclear why the rectum was not also quantified for CGRP endings in Figure 6. Quantifying rectum and distal colon would be more supportive of VMR findings because distension of the rectum and distal colon were used. There was nothing mentioned about VMR due to proximal colon distension.
- The last inconsistency relates to the electrical stimulation of the rectum to quantify the VMR in experiments 6 months post-surgery. To confirm the loss of sensory innervation after 6 months, only the distal colon (and not rectum) was quantified (Figure 5). To strengthen the authors conclusions, the CGRP and VMR studies 6-months after surgery should have been performed in the same regions. Also, experiments should have been performed TL-lesioned mice, as mentioned above. Further, using balloon distension (instead of electrical stimulation) to evoke the VMR would have been more consistent with the short-term experiments.
- Important to note here is that even though the authors state that the colorectal and bladder VMR are abolished up to 6 months after lesioning, the p value of 0.06 is usually not considered significant. There is not a statement regarding significance in the methods section. But assuming that a p value of < 0.05 is being used to determine significance, the authors need to adjust their wording when

describing these results (page 5, lines 111-113; page 9, lines 205-207).

- Finally, the authors are somewhat misleading when stating that the extrinsic sensory afferents do not influence CMMC behavior. In their experiments they measured CMMC's in an isolated colon; i.e., no extrinsic sensory neurons were intact in control (sham) or lesioned mice. It is highly likely that there would have been significant differences between sham, LS- and TL-lesioned mice if CMMC's had been measured in conditions with intact spinal circuits.

Reviewer #2 (Remarks to the Author):

The authors describe a new technique to surgically remove DRGs in live mice. The authors used this technique to interfere with communication between the rectum/colon and the brain with interesting results. I think the article is solid, it presents a new tool to understand the basis of visceral-cerebral communication. My congratulations to the authors, I think it's a good job, just a few questions:

- Do animals without DRG have any cutaneous alteration? The authors gave data on well-being (body weight, food and water consumption) and locomotion, but did not show any measures of sensitivity. Since paw sensation must follow a different pathway (mostly L4), identical paw sensation before and after the operation will show that there are no collateral alterations in the pain transmission system.

- The authors provide information on the animals operated after 6 months, it would be interesting to know if after this long period the animals continue without differences in body weight.

- A complementary video showing the process would be very helpful.

Reviewer #3 (Remarks to the Author):

Kyloh and collaborators submitted a study on a technic to remove surgically selected DRGs in order to study their implication in intestinal homeostasis. The method used in this manuscript is highly original and allows for the first time to specifically study the implication of sensory neurons in conscious and free-to-move animals. I was particularly impressed by the absence of effect of the DRG removal on the compliance. The study is very well designed and the method accurately described to allow other teams to replicate the technique. I only have minor concerns.

In addition to the fecal pellet output, it will be important to determine the impact of DRG removal on the bladder voiding in order to complete the characterization of the model.

Figure 4 L, based on the graphic, there is only two mice in the sham group which is not sufficient for a proper statistical analysis.

The quantification of the CGRP immunofluorescence in the uterine horn should be performed also in control mice in order to determine if the % in contralateral is equivalent to control.

I did not understand the link between the technique described in the study and the anterograde labelling from DRG. Could you develop it in the discussion?

Nicolas Cenac

Reviewer 1

Thank you to reviewer 1 for raising good questions and helpful comments. We appreciate the time taken to do this. We have incorporated suggestions.

We have quoted Smith-Edwards et al and Meershaert et al.

We have included a description of the sham surgery.

In Figure 4, panel M: the magenta line for the VMR in LS-lesioned mice in response to distal colon distension does not appear to represent the average of the data points at pressures 70 and 80 mmHg; the visible magenta dots are all below the line, so it is unclear what data points are causing the average to be higher? Is it possible that there were some very high values that are out of the range of the y axis?

The reviewer is correct. Well spotted. And, we add that panel Q has similar outliers. We initially limited the range to maximize spread of the individual data points to facilitate comparison but omitted to note the outliers in the original manuscript. On review, we find the extended range of the axes to accommodate the data points remains acceptable. Thus panel M and Q are revised in the new manuscript; no other graphs were changed. We thank the reviewer for bringing this to our attention.

Comment: The authors could have performed sham, LS and TL lesion surgeries to quantify long term endpoints. The long term CGRP/VMR studies (at 6 months post surgery) were simply to test whether spinal afferents could regrow from neighbouring DRG into the terminal gut. They did not. We only chose terminal large bowel for CGRP and VMR, because only this region showed abolished distension-evoked VMRs following LS DRG removal. Why the proximal colon would show regrowth, but not the LS innervated distal region is unclear. Mice lived happily for 6 months post surgery which was the important aspect of our new procedure.

Question: The reviewer asks why CGRP immunoreactivity was not quantified in the rectum, in addition to the distal colon. There is no anatomical demarcation we are aware of between the mouse distal colon from the rectum. We prefer the term colorectum, since there are not distinct sensory nerves innervating each region, there is overlap in fields between rectal and lumbar colonic (splanchnic) afferents (E.g. the rectal nerves in our anterograde tracing studies innervate the terminal 3cm of large bowel). And our DRG injection tracing results have not found evidence the rectum is different from the distal colon in terms of CGRP immunohistochemistry. In the paper, we had to come up with a distinction: “the distal colon, ~16-23 mm rostral to the anus; and rectum, ~2.5-9.5 mm from the anus” simply because the balloon was inserted different distances. Both regions can be technically stimulated during balloon distension.

“Regarding Figure 5: For consistency and more convincing results, the authors should have used the same balloon (the 7mm one) and distended the rectum and distal colon to compare pERK in various spinal regions with and without DRG lesioning. These experiments would have provided additional and more significant data on whether 'labeled line' pathways continue in the spinal cord or could provide evidence regarding the extent of spinal integration from different DRG neurons and receptive fields. “

The pERK experiments using the larger balloon were used to assess the DRG lesion surgical approach and confirm that DRG lesioning effectively attenuated the communication between DRG and spinal cord at the relevant spinal levels to where the lesions occurred. This is what these pERK studies show.

We agree further studies assess the effect of DRG lesions at different levels on pERK evoked by the smaller balloon placed rectally or the distal colon would inform spinal integration of colonic input between the spinal regions (evidence of which has been shown to occur via descending supraspinal pathways) or spinal activation relevant to receptive fields (this is what the VMR is testing).

Question: The reviewer suggests to strengthen the author's conclusions, the CGRP and VMR studies 6 months after surgery should have been performed in the same regions. As discussed above, the rectal nerves innervate the distal ~30mm of mouse large bowel. CGRP immunoreactivity is not segregated into different regions (distal colon or rectum) – which in themselves are arbitrary regions in mouse.

Question/comment: Using balloon distension instead of electrical stimulation to evoke the VMR would have been more consistent with short term experiments.

The point of electrical stimulation was to activate any mechanically sensitive or insensitive spinal afferent axons that may have regrown back after DRG removal. There was no reinnervation, which was not really surprising to us, since the cell bodies have been removed. However, it was an important experiment, especially to show that all afferents to this region (that could underlie the VMR) were abolished, not just mechanically sensitive ones. The absence of VMR to electrical stimulation was powerful supporting evidence for the consistent and prolonged loss of CGRP expressing extrinsic axons.

Regarding the statement that the reviewer focusses on a P value of 0.06 as not significant but the VMR was blocked. The VMRs were always completely blocked, in mice 6 months after LS DRG removal. We don't remember saying the data was significantly different. What we mentioned was that in all LS DRG lesioned animals there was no VMR, whereas in controls there always as. The variability in the control data led to this P value. The major result is there is no VMR when LS DRGs are removed. This has been reworded to: *“No detectable VMR was elicited in LS DRG lesioned mice, following electrical stimulation of the rectum or bladder (Supplementary Figure 3). In the same mice, VMRs to tail, hindlimb and forelimb pinches were not significantly different from controls.”*

We have added a statement in the methods, about P values for significance. We also added a paragraph on how the priors were calculated for immunohistochemical analysis, which had been inadvertently omitted.

Apologies. We have now inserted that P values <0.05 are considered significant.

Regarding extrinsic sensory afferents not affecting CMMCs. We have deleted the confusing sentence implicating spinal afferents have little role in CMMCs.

Reviewer 2:

Thank you for your time in reviewing and raising good comments. We appreciate the time devoted to this.

It's a good question, but we have not tested whether cutaneous circulation is affected after DRG removal. We hope to investigate this in the near future.

We attempted to make a video, but the procedure was very difficult to present clearly and was quite granular. Hence, we inserted the high resolution photos and cartoons to aid the readers.

Reviewer 3.

We agree. Bladder voiding requires careful investigation. We will next embark on a major study to characterise the roles of different DRG on bladder voiding. This will take a significant time to characterise in detail but is next on our list of major experiments.

In figure 4L, the reviewer was quite correct. For that data set of shams we did only have 2 controls. So, we have gone back and performed more sham experiments. The data has been recalculated and re-graphed. Well picked up by the reviewer.

We performed new experiments quantifying CGRP immunofluorescence in the uterus of control mice ($n = 5$) with the method used here, to determine if densities were equivalent those measured from the contralateral uterine horn of unilateral TL DRG-removed mice. Interestingly, we found that CGRP density was increased in the ovarian end of the contralateral uterine horn in unilateral TL DRG-removed mice compared to controls ($p = 0.01$), while the mid and cervical regions were similar between the two groups. The mechanism(s) underlying this apparent upregulation of CGRP in the contralateral uterus is presently unclear. However, this finding reiterates that the contralateral uterine horn provides an important internal control to which changes in the ipsilateral side can be compared using our unilateral DRG removal technique.

With regard to confusion between the DRG removal technique and the anterograde tracing from DRG, what we were saying in the discussion is that the same surgical technique can be used to expose and remove the DRG involves the same procedure, its just that with the anterograde tracing, the DRG are not removed from the animals. We have modified this in the discussion to read clearer.

Reviewers' comments:

Reviewer #1 (Remarks to the Author):

Thank you for addressing my comments. I would first like to emphasize my enthusiasm for the studies and results in this manuscript. I recognize the difficulty in these surgeries and commend the authors for their highly-technical skills in in vivo dissections. The manuscript in its revised form provides a novel methodology for studying spinal afferents, but still falls short of providing new scientific conclusions – an exciting finding was reported but the authors did not follow up with the anatomical experiments necessary to fully support their functional data. I have provided a few additional remarks in response to the rebuttal letter.

I appreciate the difficulty in distinguishing distal colon from rectum and agree with the authors about the absence of demarcation. However, the authors themselves made a very clear distinction between distal colon and rectum based on how far the balloon was placed when performing VMR experiments. In their words, rectal distension was applied 2.5-9.5 mm from anus and distal colon distension was applied 16-23 mm from anus, so it is unclear why these same measurements were not also used to quantify CGRP-immunoreactivity.

Further, the most interesting data in the manuscript were the VMR differences in the two regions with the surgical removal of either LS or TL DRGs and should be followed up with the proper IHC experiments. According to their VMR studies, LS afferents are mostly responsible for carrying pain information from the rectum, whereas pain signals from the distal colon are mostly carried by TL afferents. Therefore, the authors should determine whether anatomical data supports their functional findings by measuring the changes in CGRP-immunoreactivity in not only sham and LS-excised mice (as they currently present), but also in TL-excised mice (at least at early timepoints), and they should use the same length measurements used for balloon insertion to distinguish distal colon regions versus rectal regions. Based on their VMR experiments, LS but not TL removal should significantly reduce CGRP in the rectal region, whereas TL (and to a much lesser extent LS) removal should reduce CGRP in distal colon. The addition of these measurements would greatly elevate the paper and allow new scientific conclusions about differences between LS and TL afferents to be made.

Lastly, regarding the use of electrical stimulation rather than balloon distension to measure VMR 6 months post-surgery, I agree with the authors that electrical stimulation will activate both mechanically sensitive and insensitive afferents. This justification was not made clear in the paper, and electrical stimulation should have been used in conjunction with balloon distension to address the issue of silent afferents. The mechanical stimulus of pinching the tail and limbs was used for positive controls, so it is still unclear why the mechanical stimulus of distension was not also used. The problem with only using electrical stimulation to confirm the long-lasting effect of the surgery is the apparent lack of response that it evoked even in sham mice. The VMRs recorded to electrical stimulation in shams were 3, 2, and 1 $\mu\text{V}\cdot\text{s}$, whereas the average VMR in response to distension was $\sim 750 \mu\text{V}\cdot\text{s}$ (that's a 250-fold increase). Obviously, electrical stimulation does not activate as large of an area as balloon distension, explaining the tiny (and possibly negligible) VMR measurements, but it is statistically difficult to draw conclusions about a decrease in a response that is already very low to begin with (known as "floor effect"). Thus, because the electrical stimulation paradigm used by the authors did not sufficiently activate afferents in sham mice, it should not be used to make claims about the long-lasting functional effects of lesioning. Further, the authors state in their rebuttal letter and manuscript that the "entire VMR to electrical stimulation of the bladder or colorectum was still ablated" 6 months after LS lesioning. However, it appears that one LS-lesioned mouse exhibited a VMR of $\sim 0.6 \mu\text{V}\cdot\text{s}$ to rectal electrical stimulation, a value that is relatively close to those recorded in sham mice, and likely the reason that there was no significant difference between sham and LS-lesioned mice. Therefore, statistically speaking, the entire VMR was not ablated. My guess is that if balloon distension was used, the authors would have found significant differences between the two groups and provided more convincing functional evidence to go along with their anatomical evidence

about the longevity of lesioning effects.

Reviewer #2 (Remarks to the Author):

I consider the article suitable for publication in its current form. I consider the article useful and relevant for the research community.

Reviewer #3 (Remarks to the Author):

I thank the authors for their very clear responses to my comments. The results obtained on the quantification of the immunofluorescence of CGRP in the uterus of control mice are really interesting. I think they should be included in the manuscript.

Response to reviewer no.1

Regarding reviewer 1, the reviewer suggests: *“..the authors should determine whether anatomical data supports their functional findings by measuring the changes in CGRP-immunoreactivity in not only sham and LS-excised mice (as they currently present), but also in TL-excised mice (at least at early timepoints), and they should use the same length measurements used for balloon insertion to distinguish distal colon regions versus rectal regions.”*

We read the reviewer’s new additional experiment now proposed. In theory, the experiment seems logical. However, the problem with this experiment, is that this region of the colon (the distal colon) that the reviewer refers to, receives also a prominent vagal afferent supply (in contrast to the rectum where vagal afferents are absent or very much reduced in density), and it is not easy to detect or quantify any obvious loss in spinal CGRP immunoreactivity (when TL DRG are removed) from the distal colon. This is because, removal of the DRG of course does not interfere with the vagal afferents still present to this region which also express CGRP. Also, there are a few reasons why we have little confidence in how to interpret the distal colon CGRP density compared with rectal CGRP. It is like comparing apples and oranges. Importantly, the spinal afferent innervation to the rectum is much denser than the distal colon, so loss of lumbosacral DRG leads to a prominent loss in CGRP innervation compared with the distal colon. Our anterograde tracing studies from TL versus LS DRG confirms this. Secondly, as mentioned, the distal colon, unlike the rectum, receives from another extrinsic sensory innervation from the vagus, which also expresses CGRP. Finally, there is also a robust CGRP immunoreactivity in the distal colon from enteric ganglia, which is reduced in enteric ganglia in the rectum. So, removing TL DRG won’t lead to an obvious (discernable) loss in CGRP, even though the DRGs are removed. As you know, in our study, we were able to show that CGRP immunoreactivity was reduced in the rectum. This was because spinal afferents provide the major or sole extrinsic CGRP innervation to this region. And, based on the recent paper by Nestor-Kalinoski A et al. 2022. CMGH, there are much fewer myenteric ganglia in the rectum, Hence, we were able to quantify and statistically demonstrate a powerful reduction in CGRP in the rectum, when lumbosacral DRGs were removed. Based on other research in the lab, we have no confidence we could demonstrate CGRP reduction in the mid to distal colon. We are also not sure this data would change any of the conclusions. What was clear in our study was that surgical removal of TL or LS DRGs led to a reduction in function (reduced VMR).

Further supporting evidence for our concern of the experiment raised by reviewer 1 (regarding CGRP immunohistochemistry along the distal colon), is supported by similar unpublished results from our lab with CGRP immunohistochemistry in the mouse stomach. When we removed T8-T12 DRG (that supply the stomach), it was not possible to quantify a reduction in CGRP in this part of the gut, because the vagal afferents also robustly express CGRP to the stomach. So, showing a reduction in CGRP was not possible, even though we could clearly demonstrate the DRGs were removed. Again, we could only demonstrate the loss of extrinsic afferents expressing CGRP in the rectum, because only spinal afferents supply this region.

Based on our response above, even if we performed the distal colon CGRP experiment suggested by reviewer 1, we don’t know how one could interpret the data, because as

mentioned, there remains a prominent vagal afferent CGRP innervation to this region. So, to perform new surgeries, immunohistochemistry, microscopy, analysis, stats, figures and rewriting the manuscript is a lot of work, which in our view, won't add anything valuable to the reader, nor change the conclusions of this study, in any meaningful way.

Most importantly, in our study here, we functionally demonstrate that thoraco-lumbar and lumbosacral DRGs are surgically removed by the reduction or ablation in the VMR pain response to colorectal distension. Our quantification of CGRP immunoreactivity in the rectum was detailed, and this was in our view thoroughly analysed, showing clear loss of extrinsic CGRP, via ablation of spinal afferents. The point of our study was not really to compare CGRP immunoreactivity along the length of the colon (that is another whole study in itself), but rather to demonstrate a new technique to functionally ablate spinal afferents to specific spinal segments. We feel this is a major new technique with immense potential for other laboratories to address previously unresolvable questions, regarding the gut-brain axis.

Reviewer 1, at the end of their review states *"My guess is that if balloon distension was used, the authors would have found significant differences between the two groups and provided more convincing functional evidence to go along with their anatomical evidence about the longevity of lesioning effects."* Perhaps the reviewer is right. But, we feel there is an extraordinary amount of detail already in this study. Comments like: *"My guess.."* are hard to respond to.

Response to reviewer no.2

Firstly, we were delighted reviewer 2 is satisfied with our revision.

Response to reviewer no.3

Regarding the comment raised by reviewer 3 to include the new data in the manuscript, we are very happy to do this. Now, this data has been graphed (see attached) with revised figure legend and new paragraph (in red) in the manuscript (please see attached). We hope the reviewer and editorial staff feel are happy with this.

Reviewers' comments:

Reviewer #1 (Remarks to the Author):

In their first rebuttal, the authors stated that it was too difficult to quantify CGRP-immunoreactivity separately in distal colon and rectum (Figure 10), and this is why they quantified the entire colorectum, which included both regions. Therefore, I suggested that they apply their own parameters using distance from anus to separately quantify distal colon and rectum. However, now the authors' most recent response implies that they specifically quantified rectal CGRP-immunoreactivity in Figure 10. They state "As you know, in our study, we were able to show that CGRP immunoreactivity was reduced in the rectum" and "Hence, we were able to quantify and statistically demonstrate a powerful reduction in CGRP in the rectum, when lumbosacral DRGs were removed. Based on other research in the lab, we have no confidence we could demonstrate CGRP reduction in the mid to distal colon" and "Again, we could only demonstrate the loss of extrinsic afferents expressing CGRP in the rectum, because only spinal afferents supply this region." The authors' response speaks to exactly why I asked for the separate analyses – according to their data and what has been previously published, there should be a near complete abolishment of CGRP-IR in the rectum with LS lesion, and by contrast LS lesion should have a lesser impact on CGRP-IR in distal colon.

Nevertheless, after reading both rebuttals, this reviewer is now completely uncertain about what was included in the quantification shown in Figure 10, and if separate analyses of rectum and distal colon will not be provided, then the authors need to be precise about the colon region that was quantified. If only the rectum was quantified as suggested in their most recent response, then the authors should use the term "rectum" instead of "distal colon" to label graphs in Figure 10B. If both rectal and distal colon regions were included in the analysis, then the authors should clearly state that the term "distal colon" applies to both the rectum and distal colon; because the authors used rectum and distal colon separately for VMR quantification, using the term "distal colon" for CGRP-IR quantification is misleading if the rectum was also included. The readers will want to know how the regions used for CGRP staining are related to the regions used for VMR recordings to distension.

Regarding Figure 7 and the long-term "abolition" of the VMR, the authors did not respond to the most important part of the critique about the relative lack of VMR to electrical stimulation of the rectum in sham mice. There are several reasons that these data as presented cannot be used to support the authors' conclusion that the VMR (colon in particular) was abolished long term: 1. There was no statistical difference in the rectal electrical stimulation VMR in LS lesioned mice compared to sham at 6 months. 2. Compared to the VMRs evoked by distension, which was 750-fold higher, the VMRs to electrical stimulation were negligible. The authors could provide data from an unstimulated sham group as a negative control to confirm that electrical stimulation of the rectum did in fact evoke a VMR that is statistically greater than 0. Otherwise, how are the readers supposed to know that these values were more than what was recorded during baseline?

Regardless, because there were no differences (statistically) between LS lesion and sham groups, all statements about LS lesioning causing long-term abolition of rectal or colon VMR should be removed from the manuscript. In the results section (lines 156-161), the authors state that "no detectable VMR was elicited in LS DRG lesioned mice following electrical stimulation of the rectum or bladder," however in 1 of the mice, there was a VMR of ~ 0.6 - 0.7 μ Vs, and therefore this statement is not true. The authors then state that "in the same mice, VMRs to tail, hindlimb and forelimb pinches were not significantly different from controls." To be completely accurate, the authors should add here that the VMR to electrical stimulation of the rectum was also not significantly different in LS lesioned mice compared to controls. In the discussion (lines 264-267), the authors need to adjust their statement that "6 months after LS DRG removal, the entire VMR to electrical stimulation of the bladder or terminal colorectum was still ablated." If the authors can provide VMR data from unstimulated controls that show a significant increase in the VMR with electrical stimulation of the rectum, then a more accurate statement regarding the rectum would be "...was still ablated in 3 of 4, or 75%, of mice."

Although we cannot publish your paper, it may be appropriate for another journal in the Nature Portfolio. If you wish to explore the journals and transfer your manuscript please use our <https://mts-commsbio.nature.com/cgi-bin/main.plex?el=A7Cx6ENZ3C2fcc7X3A9ftdFIM0izUhPGDJ2otxj8rAQwZ> manuscript transfer portal. You will not have to re-supply manuscript metadata and files, but please note that this link can only be used once and remains active until used. For more information, please see our http://www.nature.com/authors/author_resources/transfer_manuscripts.html?WT.mc_id=EMI_NPG_1511_AUTHORTRANSF&WT.ec_id=AUTHOR manuscript transfer FAQ page.

Note that any decision to opt in to In Review at the original journal is not sent to the receiving journal on transfer. You can opt in to *[In Review](https://www.nature.com/nature-research/for-authors/in-review)* at receiving journals that support this service by choosing to modify your manuscript on transfer. In Review is available for primary research manuscript types only.

REVIEWER 1

We were puzzled reviewer 1 chose to suggest rejection at this late stage, even though in their final review they stated: *“Thank you for addressing my comments. I would first like to emphasize my enthusiasm for the studies and results in this manuscript. I recognize the difficulty in these surgeries and commend the authors for their highly-technical skills in in vivo dissections. The manuscript in its revised form provides a novel methodology for studying spinal afferents, but still falls short of providing new scientific conclusions – an exciting finding was reported but the authors did not follow up with the anatomical experiments necessary to fully support their functional data.”*

Response:

We have followed the editors suggestion. We have now incorporated additional new supplementary data figure 1, regarding CGRP immunohistochemistry of the stomach. Indeed, we had planned to publish this stomach data in another independent study (and hence this data had not been part of the original manuscript). But, to allay the concerns about the need to use caution when interpreting CGRP analysis from reviewer 1, we have included this data. The data provides direct evidence that we cannot make meaningful scientific comparisons between CGRP fluorescence intensities across different regions of the large intestine, or make meaningful comparisons about sensory neural innervation between regions. The new stomach data the editors have suggested we include, has now been included in the results section, and explained carefully as a limitation section in the discussion, as suggested.

We quantified and compared CGRP fluorescence intensity across three different parts of the stomach (corpus, fundus antrum), the lower esophageal sphincter (Les) and upper small intestine (duodenum) in 2 different cohorts of mice: (1) experimental shams and (2) mice with DRG removed bilaterally between T10-T12, which innervate the stomach and upper gut. The new figure (supplementary figure 1) shows that bilateral surgical ablation of T10-T12 DRG does not lead to a detectable reduction in CGRP immunoreactivity. The reason for this, is because in the upper gut (stomach, small bowel and upper part of colon) there is a rich vagal afferent innervation, which also expresses CGRP. And, there are significantly larger numbers of intrinsic neurons (which also express CGRP) in the proximal regions of colon, compared to distal and rectum. This is shown in Figure 3a of the recent reference by Nestor-Kalinowski A et al. (2022) *Cell Mol Gastroenterol Hepatol* 2022;13(1):309-337.e3. doi: 10.1016/j.jcmgh.2021.08.016. This figure shows there are significantly less intrinsic myenteric ganglia in the distal colon, compared with more proximal regions. Hence, quantifying CGRP immunoreactivity between regions (e.g distal colon and rectum) will, in itself, reveal major changes in intensity, independent of ablation of DRG, or any differences vagal afferent innervation.

Close inspection of the upper and lower parts of our Figure 10b, reveals that more intrinsic ganglia (CGRP+ immunoreactivity) can be seen in the proximal region (upper part of figure), compared with lower part of the figure (lower rectum), when the spinal afferents are removed from L5-S1. So, it is not an equivalent comparison to compare fluorescence intensity between these regions. Hence, why we were uncomfortable to make these comparisons, which reviewer

1 appeared to use a evidence to reject our study. The whole point of performing CGRP immunoreactivity comparisons in our manuscript, was simply to show that CGRP+ axons and terminals, can, for the first time, be selectively ablated in a peripheral organ of live mice. This, has never been performed and we feel, is a major step forward for sensory neuroscience.

Reviewer 1 question

Reviewer 1 suggests that we did not provide anatomical evidence for our functional results. This is untrue. We already showed in the first submission last year, detailed quantification in figure 5 of the significant reduction in the number neurons activated by pERK in the spinal cord, following distension of the distal colon and rectum (via thoracolumbar and lumbosacral DRGs). This data shows anatomically the major reduction in spinal neurons activated by distension of the large intestine.

Reviewer 1 question

The reviewer raises the comment: *“the term “distal colon” for CGRP-IR quantification is misleading if the rectum was also included”*.

Answer:

We accept our terminology could be misleading in Figure 10, since different laboratories have used different terms, some not even using the word “rectum”, as in the recent paper by, Nestor-Kalinowski A et al. (2022) Cell Mol Gastroenterol Hepatol 2022;13(1):309-337.e3. doi: 10.1016/j.jcmgh.2021.08.016. There is no anatomical demarcation of the rectum versus distal colon in mice. We have used the term rectum to describe where our CGRP analysis in the colon took place, consistent with our previous terminology. To clarify, we have in this revision now explained and define more clearly the rectum region from the distal colon in the methods, in terms of CGRP analysis.

Question from reviewer 1:

Reviewer asks: *“Nevertheless, after reading both rebuttals, this reviewer is now completely uncertain about what was included in the quantification shown in Figure 10, and if separate analyses of rectum and distal colon will not be provided, then the authors need to be precise about the colon region that was quantified”*

Answer:

Apologies for the confusion. As mentioned before, in reality, in nature, there is no clear anatomical demarcation identifying in the large intestine (of mice) any region that segregates the rectum from colon. We use the term rectum in mouse, some don't – like the recent paper by Brian Davis's laboratory with Marthe Howard (mentioned above). We cite this paper now to clarify this. The important thing, is defining what the rectum is, so future scientists can understand what was performed. For clarity, we have, on page 16 of this revision, now stated *“For analysis of CGRP, we defined the rectal region as the region predominantly innervated by*

the rectal nerves (within 16mm of the anal sphincter).” The figure has been appropriately relabelled to use the term rectum (for CGRP) analysis, not colon. We explain in the manuscript, that our use of the term rectum is defined by the region of large bowel, predominantly innervated by the rectal (not splanchnic) nerves.

Question from Reviewer 1: makes a new comment in the 2nd resubmission, that wasn't made in the first submission regarding data in Figure 7, which is what a decision to reject appears to be made on. They state: “*Regardless, because there were no differences (statistically) between LS lesion and sham groups, all statements about LS lesioning causing long-term abolition of rectal or colon VMR should be removed from the manuscript.*”

Response.

The suggestions and advice of reviewer 1 about the interpretation of our P values (i.e. $P=0.06$) is the exact opposite of the suggestions made to scientists by the journal **Nature** (see evidence below). Reviewer 1 now states because one data set (in Figure 7) arrived at an overall $P=0.06$, we now need to change our conclusions and state that the VMR response was not blocked, 6 months after DRG surgery. We looked over all the recordings again to confirm; and the VMR is completely ablated 6 months after DRG removal. *Nature* published not long ago a specific comment addressing P values in their articles. They state: “**Let's be clear about what must stop: we should never conclude there is 'no difference' or 'no association' just because a P value is larger than a threshold such as 0.05 or, equivalently, because a confidence interval includes zero.**” This article was very widely appreciated and followed by top scientists. Nature's article was published here: <https://www.nature.com/articles/d41586-019-00857-9>. Based on our actual findings that the VMR was blocked in all mice, and *Nature's* viewpoint, we feel it is completely unjustified to follow the suggestion of Reviewer 1 and now change our conclusions. It is very disappointing to us that of three reviewers, this comment of reviewer 1 appears to have had a major impact on a decision to reject outright our entire manuscript after 9 months of corrections and resubmissions. The 1 animal in Figure 7 (left hand graph) that the reviewer has now singled (but didn't mention in the first reviewer's comments), is totally within the electrode noise. All EMG recordings have some degree of electrical noise. So, reviewer 1 has singled out one data point, in one graph on Figure 7 (which has a P value of $P=0.06$) and used this a major reason to reject the whole manuscript. The important result was that all mice showed complete abolition of VMR, 6 months ago DRG removal. Unsurprisingly, based on this, we do not feel comfortable completely revising of our interpretation that the important findings and the whole manuscript. We hope you agree. And this is a major result of the study, which will serve a key foundation for future gut-brain research.

Reviewers' comments:

Reviewer #4 (Remarks to the Author):

This is a focused review of a revised manuscript from Kyooh et al. that describes a surgical approach to ablation of DRG function at specific spinal levels and then uses this approach to test the relative effects of thoracolumbar vs. lumbosacral DRG removal on visceromotor reflexes elicited by intraluminal distention of the distal colon or rectum in adult mice. In particular, this reviewer was asked to comment on the authors' rebuttal to Reviewer 1 feedback on the second revised version of the manuscript. There are two main points of friction to consider:

Figure 7 – Reviewer 1 suggests that no conclusions can be drawn from the first 2 panels in this figure. The authors appropriately challenge this feedback – while the sample size is small, these are technically challenging and heroic experiments and the data do support the authors' conclusion that there is no detectable response to electrical stimulation in the LS lesioned mice. As the reviewer suggests, the optimal experiment would have been to test VMR to intraluminal distention, as was done at the 2 week timepoint. The absence of this experiment, however, does not invalidate the authors findings but this could be identified as a limitation or future experiment.

Figure 10 – The observations on long-term depletion of CGRP in a region-specific manner are interesting but Panels B & D are very difficult to understand. It is not clear what exactly was quantified – for example, what do "posterior density" and "proportion of foreground pixels" refer to? What are the dot colors representing? No color legend for these data points is evident. The experiment and conclusions drawn seem reasonable but the quantification is very unclear. Also, given the challenge of assessing intrinsic vs. extrinsic CGRP, it would be helpful if the authors could provide high magnification insets of what the intrinsic signal looks like in rectum vs. more proximal colon.

Minor comment on Figure 1 – the schematics are elegant and very helpful. For those readers who are not immediately familiar with the complexities of PNS neuroanatomy, it would be helpful to add identifying information to the call-out bubble with resected area magnified (ie. SC for spinal cord, DRG, PG, etc).

Response to reviewer no.4.

Thank you very much for the excellent points. We have incorporated all the suggested changes.

Regarding figure 7, we have modified the discussion to acknowledge the $P=0.06$ value and the meaning of this, even though all VMR responses were blocked 6 months after DRG surgery. We mentioned in the discussion the lack of distension applied to the rectum at 6 months could be considered a limitation of our study.

We appreciate the description of the CGRP analysis was complex and not simple to understand. We have reworded this (see in red) and explained in more detail how this analysis was performed and what the axes refer to with greater clarity. Fair enough point.

Regarding figure 10, we added a new supplementary figure 2 which shows in high magnification the changes in neurochemistry of CGRP after DRG lesion, which was specifically requested. We feel this was a helpful improvement and thanks for asking for this.

We have modified figure 1 to include more names of structures in the diagram to improve clarity, and updated the legend to include the new terms used to describe the anatomy. Good point !